# Technical note: Stochastic simulation of streamflow time series using phase randomization

Manuela I. Brunner[1], András Bárdossy[2], and Reinhard Furrer[3]

[1]Mountain Hydrology and Mass Movements, Swiss Federal Institute for Forest, Snow and Landscape Research WSL, Birmensdorf ZH, Switzerland
[2]Institute for Modelling Hydraulic and Environmental Systems, University of Stuttgart, Stuttgart, Germany
[3]Institute of Mathematics, University of Zurich, Zurich, Switzerland

Correspondence: Manuela Brunner (manuela.brunner@wsl.ch)

**Abstract.** Stochastically generated streamflow time series are widely used in water resource planning and management. Such series represent sets of plausible yet unobserved streamflow realizations which should reproduce the main characteristics of observed data. These characteristics include the distribution of daily streamflow values and their temporal correlation as expressed by short- and long-range dependence. Existing streamflow generation approaches have mainly focused on the time domain, even though simulation in the frequency domain provides good properties. These properties comprise the simulation of both short- and long-range dependence, as well as extension to multiple sites. Simulation in the frequency domain is based on the randomization of the phases of the Fourier transformation. We here combine phase randomization simulation with a flexible, four-parameter kappa distribution, which allows for the extrapolation to yet unobserved low and high flows. The simulation approach consists of seven steps: 1) fitting the theoretical kappa distribution, 2) normalization and deseasonalization of the marginal distribution, 3) Fourier transformation, 4) random phase generation, 5) inverse Fourier transformation, 6) back transformation, and 7) simulation. The simulation approach is applicable both to individual and multiple sites. It was applied to and validated on a set of four catchments in Switzerland. Our results show that the stochastic streamflow generator based on phase randomization produces realistic streamflow time series with respect to distributional properties and temporal correlation. However, cross-correlation among sites was in some cases found to be underestimated. The approach can be recommended as a flexible tool for various applications such as the dimensioning of reservoirs or the assessment of drought persistence.

**Keywords**: Fourier transformation, spectral analysis, generator, frequency domain, catchment, temporal dependence, correlation

**Keypoints**:

1. Stochastic simulation of streamflow time series for individual and multiple sites by combining phase randomization and the kappa distribution

2. Simulated time series reproduce temporal correlation, seasonal distributions, and extremes of observed time series

3. Simulation procedure suitable for use in water resource planning and management

# 1 Introduction

Stochastically generated streamflow time series are used in various applications of water resource planning and management. These applications include water and reservoir management, the determination of the dimensions of hydraulic structures such as reservoirs, and the estimation of hydrological extremes such as droughts and floods. Stochastically generated time series mimic the characteristics of observed data and represent sets of plausible realizations of streamflow sequences (Ilich, 2014; Borgomeo et al., 2015; Tsoukalas et al., 2018b). They are essential for many uncertainty studies in hydrology because they can serve as input for deterministic water system models in which they allow for the propagation of natural variability and uncertainty (Tsoukalas et al., 2018b).

Stochastic models for the generation of synthetic streamflow time series need to fulfill certain requirements. They should reproduce both the marginal distribution of observed streamflow time series as well as their temporal dependence structure (Sharma et al., 1997; Salas and Lee, 2010). Temporal dependence encompasses both short- and long-range dependence. While short-range dependence typically refers to the dependence of daily streamflow values measured within a few days, long-range dependence refers to dependencies across months or years. This temporal dependence has been found to depend on magnitude, in that low values have stronger dependence than high values (Lee and Salas, 2011). A proper representation of this long-range dependence is of particular importance in studies where storage in reservoirs is of interest (Tsoukalas et al., 2018b). If one is interested in extreme events, the model should allow for the generation of values that go beyond the magnitude of those observed (Herman et al., 2016). This requires the choice of a suitable theoretical marginal distribution. Streamflow typically exhibits a skewed distribution, which requires the use of a three or more-than-three-parameter distribution (Koutsoyiannis, 2000; Blum et al., 2017). Studies looking at individual hydrological events such as floods or droughts require a daily resolution. Therefore, the stochastic model should allow for outputs at such a fine temporal resolution. Often, study regions encompass several sites whose streamflows are correlated. Consequently, the model ideally not only allows for the simulation of streamflow at individual sites, but also for the joint simulation of streamflow at multiple sites, taking into account their spatio-temporal dependence. From a practitioner's point of view, the model should not only reproduce the characteristics of the observed data, but it should also be simple (Sharma et al., 1997).

Many different approaches have been proposed for the stochastic simulation of streamflow time series, each able to fulfill some but usually not all of the desired properties listed above. One commonly used approach is the use of a synthetic weather generator in combination with a rainfall-runoff model (Pender et al., 2015). This approach is affected by uncertainties due to hydrological model selection and calibration, which can be avoided by using direct synthetic streamflow generation approaches (Herman et al., 2016). According to Stedinger and Taylor (1982), the development of such direct approaches consists of the following steps: 1) selection of a model for seasonal marginal distributions, 2) selection of a model for spatial and temporal dependence, and 3) validation of the model. Different groups of direct approaches exist which are distinct in terms of their flexibility regarding marginal distributions and temporal dependence structures.

A first group of models consists of parametric models such as autoregressive moving average (ARMA) models and their modifications. While these models are commonly used in stochastic hydrology, they only allow for modeling short-range

dependence because their autocorrelation decreases strongly with increasing lag time (Sharma et al., 1997). This means they guarantee neither the reproduction of observed persistence of annual flows nor the correlation structure among flows in different months (Stedinger and Taylor, 1982). This makes them unsuitable for applications where long-range dependence is important (Koutsoyiannis, 2000). However, AR models can be used to generate seemingly long-memory processes if a parametric auto-

correlation structure is used to fit the data (Papalexiou, 2018). A second group of parametric models is based on the temporal disaggregation of annual series and enables the representation of long-range dependence (Stedinger and Taylor, 1982; Salas and Lee, 2010). These models include fractional Gaussian noise models (Mandelbrot, 1965), fast fractional Gaussian noise models (Mandelbrot, 1971), broken line models (Mejia et al., 1972), and fractional autoregressive integrated moving average models (Hosking, 1984). Disaggregation models can be extended to multi-site applications (Grygier and Stedinger, 1988). However,

this group of models has been shown to exhibit parameter estimation problems, and only allows for the representation of a narrow range of autocorrelation functions (Koutsoyiannis, 2000). A third group of models is nonparametric in its approach and includes kernel density estimation (Lall and Sharma, 1996; Sharma et al., 1997) and various bootstrap approaches. The latter include simple bootstrap, which is only useful if data are uncorrelated, moving block-bootstrap, nearest-neighbor bootstrap (Salas and Lee, 2010; Herman et al., 2016), matched-block bootstrap (Srinivas and Srinivasan, 2006), and maximum-entropy

bootstrap (Srivastav and Simonovic, 2014), which also take lagged correlations into account. These nonparametric techniques resample from the data with perturbations and directly reproduce the characteristics of the original data (Sharma et al., 1997). However, the reproduction of long-range dependence is difficult and variance can be under- or overestimated (Salas and Lee, 2010). To allow for values that go beyond the observed distribution, Salas and Lee (2010) proposed a model employing $k$-nearest neighbor resampling with a gamma kernel perturbation. A further group of models consists of models that employ

Markov chains and their variations. These models account for transition probabilities between different hydrological states (Stagge and Moglen, 2013; Bracken et al., 2014; Pender et al., 2015) and can be combined with nonparametric approaches such as $k$-nearest neighbors (Prairie et al., 2008). They can be extended to multiple sites by scaling the simulated values at individual sites with spatially correlated random numbers (Mehrotra and Sharma, 2006).

Several alternatives to these well-established simulation procedures have been proposed, which allow for a flexible choice

of marginal distributions. These include models where the temporal dependence structure is modelled with copula functions, which are, however, difficult to apply for higher orders of autocorrelation (Lee and Salas, 2011). Examples of new simulation procedures based on the Autoregressive to Anything (ARTA) model proposed by Cario and Nelson (1996) are the SMARTA model by Tsoukalas et al. (2018b) or the SPARTA model by Tsoukalas et al. (2018a), which employ Nataf's joint distribution model for the simulation of stochastic time series, representing both short- and long-range dependence. In addition, simulation

schemes based on wavelet decomposition, which avoid assumptions about the temporal dependence structure, have been proposed by Kwon et al. (2007); Wang et al. (2010) and Erkyihun et al. (2016). Borgomeo et al. (2015) have shown how simulated annealing can be used to generate synthetic streamflow time sequences that represent possible climate-induced changes in user-specified streamflow properties.

All these previously mentioned models are based on the time domain. An alternative to time-domain models is frequency-

domain models (Shumway and Stoffer, 2017), which allow for the simulation of surrogate data with the same Fourier spectra

as the raw data (Theiler et al., 1992). Such methods are based on the randomization of the phases of the Fourier transformation and have been commonly applied in hypothesis testing, when identifying nonlinearity in time series (Schmitz and Schreiber, 1996; Kugiumtzis, 1999; Venema et al., 2006; Maiwald et al., 2008), and in trend detection (Radziejewski et al., 2000). We hereafter refer to such methods, which are also known as amplitude-adjusted Fourier transformations (AAFT) (Lancaster et al., 2018), as *phase randomization* simulations. Serinaldi and Lombardo (2017) used an iterative AAFT method to generate binary series of rainfall occurrence and non-occurrence. An extension of the amplitude-adjusted Fourier transformation has been presented by Keylock (2007) who employed randomization procedures to wavelet decomposed signals to generate surrogate data. In hydrology, phase randomization simulation has rarely been applied for purposes other than hypothesis testing (Fleming et al., 2002) even though it has desirable properties which make it suitable for a wider range of applications. Indeed, its implementation is relatively simple, it can simulate time series with both short- and long-range dependence, and it can be extended to multiple sites. However, its application is often limited to the reproduction of the empirical distribution of the data. We here propose the use of phase randomization simulation for the stochastic generation of streamflow time series at individual and multiple sites. To allow for non-empirical distributions, we combine the data simulated by phase randomization with the flexible, four-parameter kappa distribution introduced by Hosking (1994) as a generalization of the three-parameter kappa distribution suggested by Mielke (1973). The stochastic streamflow generation approach shall represent a flexible tool, which is easy to apply, and generalizable to different contexts. This is enabled by combining a nonparametric time dependence model with a flexible four-parameter distribution. The simulation approach can be tailored to the specific problem at hand and be used for various water resource management applications.

We now turn to some theoretical background on Fourier transformation and phase randomization. For a more detailed introduction to the Fourier transformation, the reader is referred to textbooks by Morrison (1994) or Shumway and Stoffer (2017). We then discuss the use of phase randomization for the stochastic generation of streamflow time series. For illustration purposes, we apply and validate the approach on a set of four catchments in Switzerland. Finally, we discuss potential applications of the simulation approach.

## 2   Theoretical background

The basic idea behind all surrogate methods is to randomize the Fourier phases of the underlying (hydrological) process. The Fourier transformation converts a time-domain signal into a frequency-domain signal, which is complex-valued. This transformation may be depicted as a decomposition of the time series into sine and cosine waves of different amplitude, phase, and period (Fleming et al., 2002; Shumway and Stoffer, 2017). In the frequency domain, the power spectral density (power spectrum) expresses the same information in cycles as the autocovariance function expresses in lags in the time domain. The periodogram, the empirical counterpart of the power spectrum, shows high values at those frequencies which correspond to strong periodic components (Shumway and Stoffer, 2017).

The surrogate approach utilizes the property that realizations of linear Gaussian processes differ only in their Fourier phases and not their power spectrum. It preserves the autocorrelation structure of the raw series by conserving its power spectrum

through phase randomization. The procedure consists of three main steps (Radziejewski et al., 2000; Maiwald et al., 2008; Kim et al., 2010). In the first step, the streamflow series is converted from the time domain to the frequency domain by Fourier transformation (Morrison, 1994). In this frequency domain, the data are represented by the phase angle and the power spectrum, as represented by the periodogram. The phase angle $\theta$ of the power spectrum is uniformly distributed over the range of $-\pi$ to $\pi$. In the second step, the phases in the phase spectrum are randomized while the power spectrum is preserved. In the third step, the inverse Fourier transformation is applied to transform the data from the frequency domain back to the temporal domain (Maiwald et al., 2008).

The Fourier transformation of a given time series $x = (x_1, \ldots, x_t, \ldots, x_n)$ of length $n$ is

$$f(\omega) = \frac{1}{\sqrt{2\pi n}} \sum_{t=1}^{n} e^{-i\omega t} x_t, -\pi \le \omega \le \pi, \tag{1}$$

where $i = \sqrt{-1}$ is the imaginary unit. The original time series can be recovered by the back transformation

$$x_t = \sqrt{\frac{2\pi}{n}} \sum_{j=1}^{n} e^{i\omega_j t} f(\omega_j), t = 1, 2, \ldots, n, \tag{2}$$

if the transformation is calculated for discrete frequencies $\omega_j = 2\pi/n, j = 1, 2, \ldots, n$. The Fourier transformation surrogate method constructs a new time series $y_t$ with the same periodogram as the observations. Apart from this, the new series are statistically independent of $x_t$. This can be achieved by fixing the Fourier amplitudes $|f(\omega_j)|$ and replacing the Fourier phases $\phi(\omega_j) = \arg(f(\omega_j))$ by uniformly distributed random numbers $\phi_{\text{rand}}(\omega_j) \in [-\pi, \pi]$. A new realization is given by

$$y_t = \sqrt{\frac{2\pi}{n}} \sum_{j=1}^{n} e^{i\omega_j t} |f(\omega_j)| e^{i\phi_{\text{rand}}(\omega_j)}. \tag{3}$$

The surrogate data consist of the same values as the original data in another temporal order but with the same time dependence structure as the original data (Schreiber and Schmitz, 2000). The approach can be extended to multiple sites by multiplying the phases of each site by the same set of random phases. This is possible because the cross-spectrum, which describes the cross-correlation of the data in the frequency domain, reflects relative phases only (Prichard and Theiler, 1994; Schreiber and Schmitz, 2000).

## 3 Methods

### 3.1 Stochastic streamflow simulation

Here, we use phase randomization to simulate stochastic streamflow time series to be used in various water resource management studies. The stochastic series generated using phase randomization are combined with a theoretical distribution to allow extrapolation to unobserved values, which still realistically represent daily streamflow values. The observed streamflow time series require pre-treatment before phase randomization can be applied. First, they need to be normalized because phase randomization assumes Gaussianity (Maiwald et al., 2008). Second, they need to be deseasonalized in order to remove monthly/daily fluctuations (Pender et al., 2015). The stochastic simulation procedure consists of the following seven steps.

1. **Fitting of theoretical kappa distribution:** The four-parameter kappa distribution (Hosking, 1994) is fit to the daily values of the observed input time series using L-moments. This distribution will be used for the back transformation in Step 7, and permits extreme values going beyond the empirical distribution to be obtained. It has four parameters and its cumulative distribution function is expressed as

$$F(x) = \left\{ 1 - h[1 - k(x - \xi)/\alpha]^{1/k} \right\}^{1/h}, \tag{4}$$

where $\xi$ is the location parameter, $\alpha$ is the scale parameter which must be positive, and $k$ and $h$ are the shape parameters.

The kappa distribution was found suitable for fitting observed streamflow data in U.S. catchments (Blum et al., 2017). A suitable fit was also found for our data as confirmed by the Kolmogorov–Smirnov and Anderson–Darling tests which did not reject the null hypothesis at $\alpha = 0.05$ for most catchments. We fit a separate distribution for each day to take into account seasonal differences in the distribution of daily streamflow values. To do so, we used the daily values in a 30-day window around the day of interest. This procedure guarantees a large enough sample for the parameter fitting procedure, and allows for smoothly changing distributions along the year. For leap years, flows from February 29 were removed to maintain constant sample sizes across years as in Blum et al. (2017).

2. **Normalization and deseasonalization of the marginal distribution**: The input time series are normalized using the normal transform, i.e., values corresponding to a certain rank are replaced with respective values from a standard normal distribution. The normal transform is applied to each day of the year separately, which results in the deseasonalization of the marginal distribution of the data.

3. **Fourier transformation:** The normalized and deseasonalized data are transferred to the frequency domain using the Fourier transformation (Equation 1). The Fourier phases (i.e., the arguments of the Fourier transformation) are computed.

4. **Random phase generation:** Random phase series are generated by sampling from the uniform phase distribution. The observed spectrum (i.e., the modulus of the Fourier transformation) is preserved.

5. **Inverse Fourier transformation:** The random phases are combined with the observed spectrum and inverse Fourier transformation is applied (Equation 2) to transform the data back to the time domain.

6. **Back transformation:** The data are back transformed from the normal to the kappa domain using the fitted daily kappa distributions (Equation 4), which achieves reseasonalization. This is done by generating a sample of length $n$ (length of observed time series) and reassigning values according to the ranks in the simulated series. Negative simulated values are replaced by values sampled from a uniform distribution in the interval [0, min($x$)], where min($x$) represents the minimum of the observed values corresponding to the day under consideration. Using the empirical distribution instead of the kappa distribution would prevent us from obtaining values that go beyond the range of observed data (Srinivas and Srinivasan, 2006). Depending on the input time series, other suitable theoretical distributions than the kappa distributions could be used for back transformation.

7. **Simulation:** Steps 4-7 are repeated $m$ times to generate $m$ time series of the same length as the observed time series.

The method is extended to the simulation of stochastic streamflow time series at multiple sites. To model the cross-correlation between sites, the phase randomization performed in Step 4 of the procedure is performed in the same way for all the stations in the dataset (Prichard and Theiler, 1994). In contrast, the parameters of the monthly kappa distributions and the power spectrum are calculated for each individual site separately.

## 3.2 Model validation

The simulation was validated on the observed streamflow time series of a set of four catchments in Switzerland (Figure 1), namely, Plessur-Chur, Birse-Moutier, Thur-Jonschwil, and Cassarate-Pregassona. The catchments are characterized by diverse catchment characteristics and flow regimes (Table 1). Their catchment areas range between 74 and 493 km² and their mean elevations between 930 and 1,850 m a.s.l. Plessur represents a catchment with a melt-dominated flow regime with high flows in summer but low flows in winter. In contrast, the flow regimes of Birse and Thur are dominated by precipitation with high flows in winter and low flows in summer. The regime of Cassarate shows two peaks, one in spring due to melt processes, and one in fall due to precipitation.

The model outlined in the previous section was fit to the observed time series over 50 years (1960-2009) for each individual catchment. The application of this approach is only recommended for records longer than 30 years to reduce uncertainty in the estimation of the parameters of the kappa distribution. The model was then run, on the one hand, for each individual catchment and, on the other hand, for the four sites jointly. In both cases, 100 sets of stochastic streamflow time series of the same length as the observed series were generated as in Salas and Lee (2010) and Pender et al. (2015).

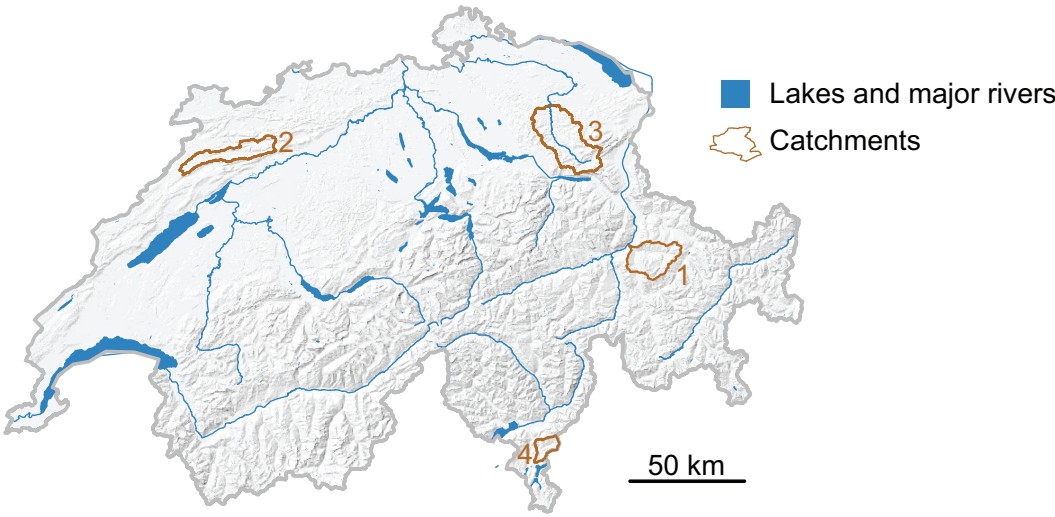

**Figure 1.** Map showing the four Swiss catchments: 1) Plessur, 2) Birse, 3) Thur, and 4) Cassarate.

**Table 1.** List of catchments and catchment summary including ID, river name, gauging station, catchment area, station elevation, mean elevation, and flow regime.

| ID | River | Gauging station | Area (km$^2$) | Station elevation (m a.s.l.) | Mean elevation (m a.s.l.) | Flow regime |
|----|-------|-----------------|---------------|------------------------------|---------------------------|-------------|
| 1 | Plessur | Chur | 263 | 573 | 1,850 | Melt-dominated |
| 2 | Birse | Moutier | 183 | 519 | 930 | Rainfall-dominated |
| 3 | Thur | Jonschwil | 493 | 534 | 1,030 | Rainfall-dominated |
| 4 | Cassarate | Pregassona | 74 | 291 | 990 | Mixed |

Both the temporal correlation structure and seasonal streamflow statistics were used to compare observed and simulated streamflow time series in order to assess the validity of the stochastic streamflow generation model. As in Kim et al. (2010), we used the autocorrelation function on daily values to represent the short-range temporal correlation. Further, we also used the partial autocorrelation function (Stedinger and Taylor, 1982). In addition to short-range dependence, long-range depen-
dence was assessed by looking at the autocorrelation function of annual discharge sums. The seasonal statistics were validated with respect to the seasonal distributions (winter: Dec–Feb, spring: Mar–May, summer: Jun–Aug, and fall: Sep–Nov) and the monthly means, maxima, minima, and standard deviations. In addition to general distribution characteristics, the approach was validated for low and high flows because these characteristics are often of interest in hydrological simulation studies (Borgomeo et al., 2015). High and low flows were defined as above or below threshold values, respectively. For high flows, the 95[th]
percentile was used as a threshold, while the 5[th] percentile was used for low flows.

## 4   Results

### 4.1   Simulation at individual sites

The stochastic streamflow generator was found to produce realistic annual hydrograph realizations as illustrated in Figure 2 for the Plessur catchment (Figure 1). This is confirmed visually by observing the temporal correlation structure, as well as the
seasonal statistics (see Figure 3 for Thur and Plessur).

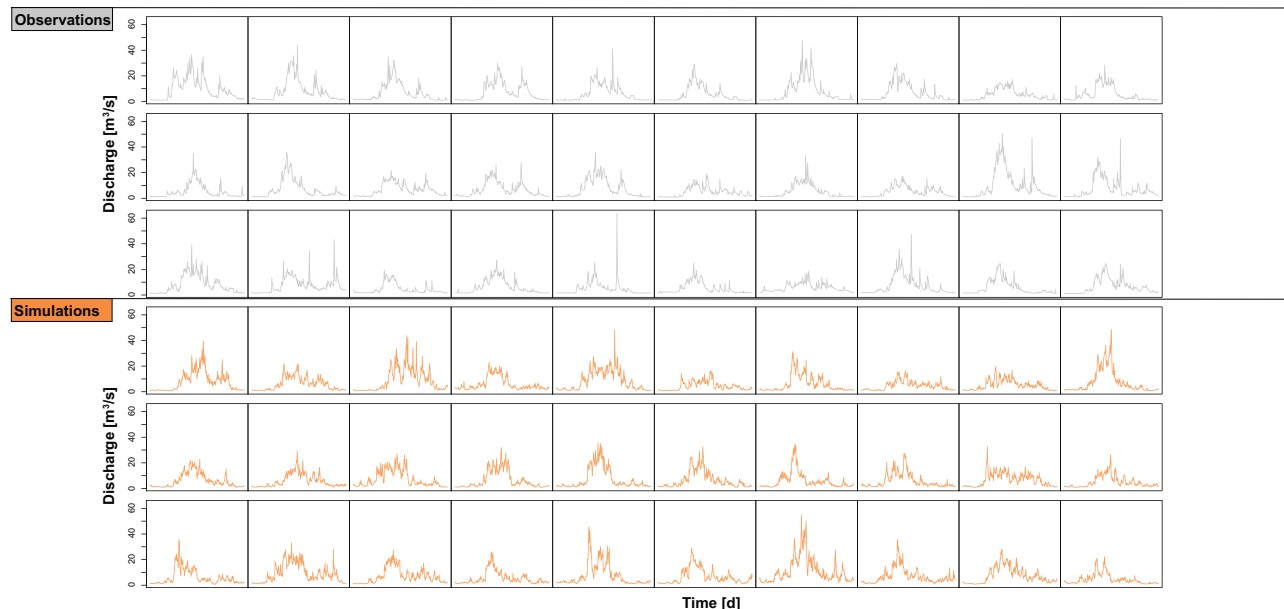

**Figure 2.** Observed (grey) and stochastically generated (orange) annual hydrographs at daily resolution over 30 years for the Plessur catchment.

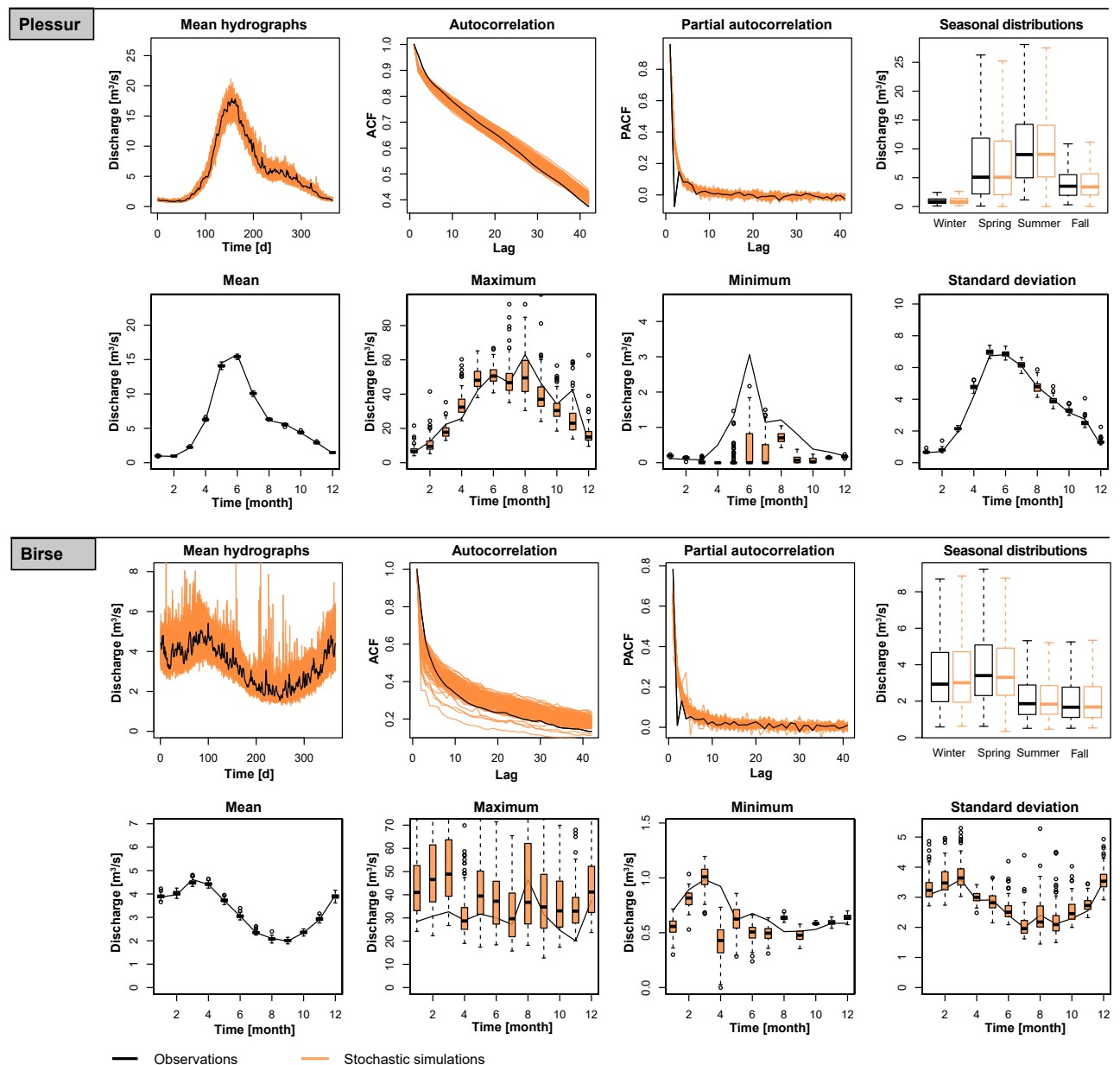

**Figure 3.** Comparison of observed and stochastically generated time series for the melt-dominated Plessur catchment (upper panels) and the rainfall-dominated Birse catchment (lower panels) for the following characteristics: Mean hydrograph over 50 years, autocorrelation function, partial autocorrelation function, seasonal distributions, monthly means, monthly maxima, monthly minima, and monthly standard deviations. Black lines represent observations while orange lines represent simulations.

The stochastic generator produces time series with mean regimes similar to the observed mean regime, and reproduces both the autocorrelation (ACF) and partial autocorrelation functions (PACF). Seasonal distributions match well thanks to the good

fit of the kappa distribution to the data. Monthly means and standard deviations match particularly well, while monthly maxima and minima show some deviations from the observed maxima and minima, as was intended by using a theoretical instead of an empirical distribution. The suitability of the kappa distribution to produce realistic high and low flows is confirmed in Figure 4. The distribution produces low flows similar to observed low flows but with different outliers. In two catchments (Thur and Cassarate) however, observed low flows were rather overestimated. High flow distributions match well in all catchments, and values exceeding observed values are generated. The four-parameter kappa distribution (Houghton, 1978; Griffiths, 1989) was found to be more suitable for representing daily streamflow values compared to distributions with even more parameters, which are rather prone to over-fitting. Similarly, tests on distributions with only three parameters (e.g. Burr type XII (Burr, 1942) and generalized Gamma distributions (Stacy and Mihram, 1965)) were here not satisfactory because the distributions were not flexible enough. In cases, where distributions with less parameters provide a satisfactory fit, they could, however, be used instead of the kappa distribution to ensure model parsimony.

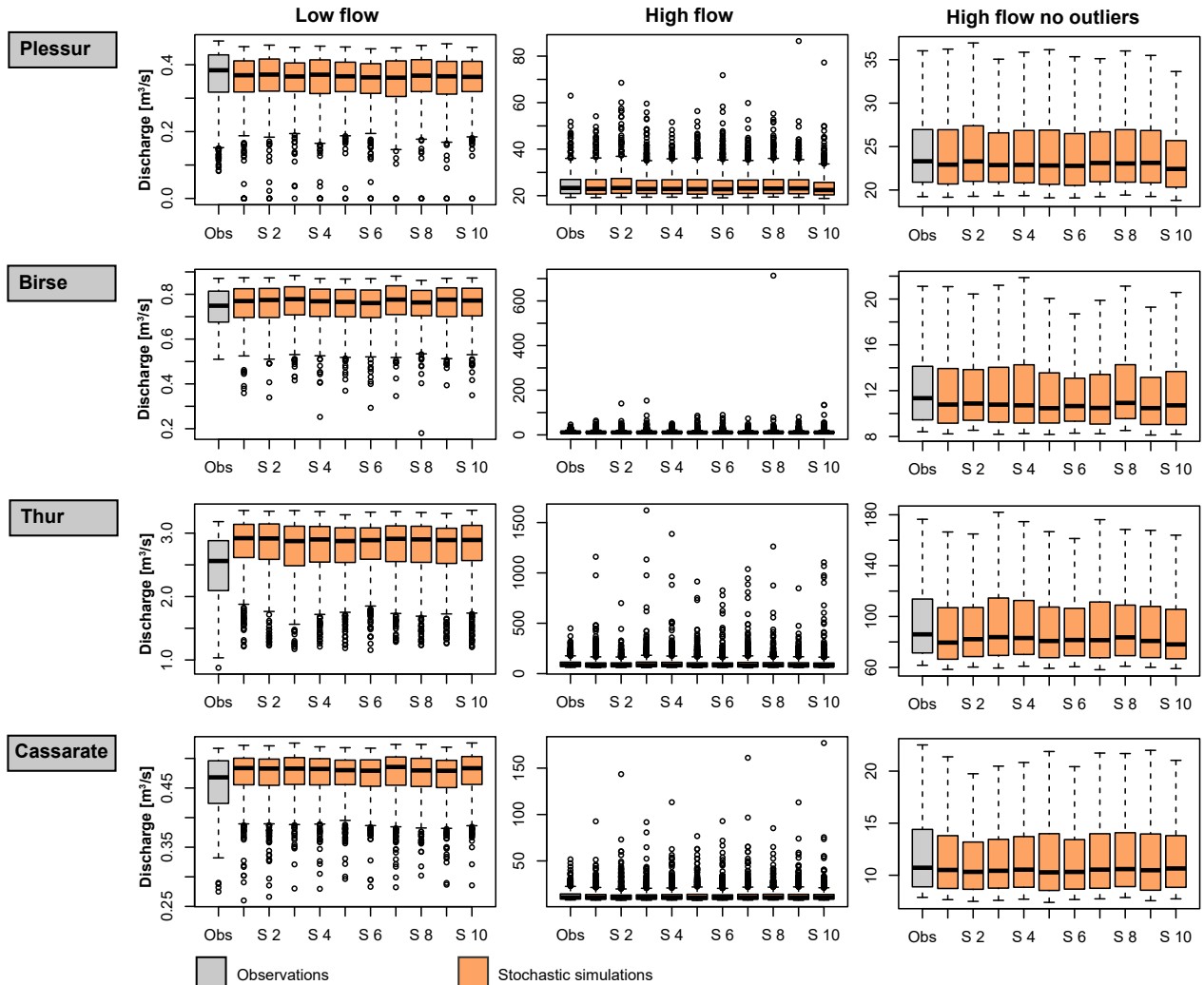

**Figure 4.** Low and high flows for observed (grey) and simulated (orange) time series for the four catchments Plessur, Birse, Thur, and Cassarate. The results are given for ten simulation runs (S1-S10), and high flows are plotted with (middle panel) and without outliers (right panel). Whiskers extend to the lowest/highest data point which is still within 1.5 times the interquartile range.

The stochastic streamflow generator is not only able to reproduce the streamflow distribution and the short-range dependence in the data, but also the long-range dependence over several years (Figure 5). Both the rapid decrease in the ACF at short lags (up to five years) and the cyclical behavior at lags longer than five years are reproduced as well.

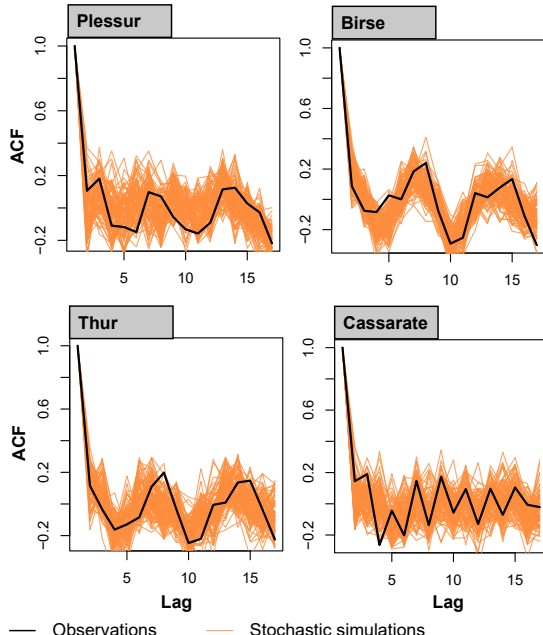

**Figure 5.** Autocorrelation (ACF) of annual streamflow sums of the observed and simulated streamflow time series for the catchments Plessur, Birse, Thur, and Cassarate.

The good performance of the stochastic streamflow generator with respect to streamflow distribution and temporal correlation - both short and long range - is not limited to these four example catchments but generalizes to other data sets used as input.

## 4.2  Simulation at multiple sites

The stochastic streamflow generator can be extended from the simulation at individual sites to the joint simulation at multiple sites. In addition to reproducing distribution and temporal correlation at individual sites, it should then be able to reproduce the cross-correlation among sites, which describes the similarity of time series at two sites. Figure 6 shows the cross-correlation function (CCF) for pairs of stations among the example catchments for the observed time series and the 100 simulation runs. Cross-correlation is already generally low for observations because the selected sample catchments are characterized by di-

verse discharge regimes and seasonality. The shape of the cross-correlation is reproduced for all pairs of stations. However, the magnitude of cross-correlation is underestimated for certain pairs of stations in the simulated time series compared to the observed series independently of the simulation run considered. For the catchment pair Birse-Thur, whose discharge behavior is rainfall-dominated, the simulated cross-correlation is much lower than the observed one. In the observations, spatially consistent rainfall events lead to a joint rise in discharge at both stations. This behavior is not captured by the stochastic dis-

charge generator. The underestimation of cross-correlation is also visible when looking at the cross-correlation of below- or above-threshold events (not shown).

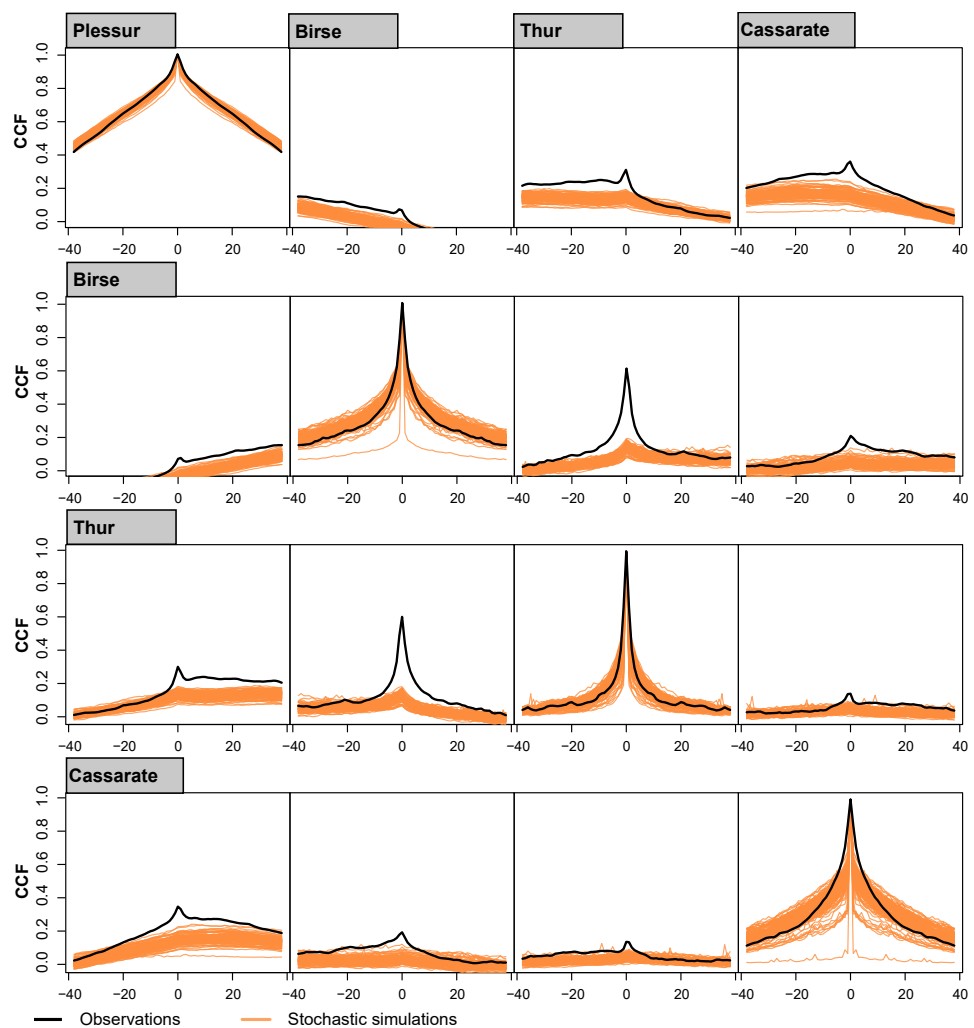

**Figure 6.** Cross-correlation function (CCF) of observed (black line) and simulated (orange lines) daily streamflow for pairs of stations at Plessur, Birse, Thur, and Cassarate.

## 5    Discussion and Conclusions

The stochastic streamflow generator based on phase randomization has been shown to produce realistic streamflow time series with respect to both distributional properties and temporal correlation. Compared to models commonly used for the stochastic generation of streamflow time series, such as autoregressive moving average models, the simulation approach presented here

5    not only reproduces short-range but also long-range dependence. However, the representation of this dependence is limited to ranges within the length of the observed time series. Instead of producing one long time series, the simulation procedure allows for the simulation of multiple series of the same length as the original series. The use of ensembles of the same length of the observed time series might not be equivalent to using a long time series. Still, long-range dependence features may not

be generated in either case since the model is fitted based on a limited number of years of observations. While the reproduction of the temporal dependence was well reproduced here, this is not necessarily the case under all conditions. Embrechts et al. (2010) have shown that any nonlinear transformation of a Gaussian time series, which is done during backtransformation, reduces the strength of the linear correlations in the time series as expressed by Pearson's correlation coefficient and preserves

only rank correlations. If one is working with heavy-tailed and zero inflated marginals (as present when looking at intermittent processes), it can happen that autocorrelations are reduced during backtransformation (Papalexiou, 2018).

    Phase randomization was here combined with the flexible four-parameter kappa distribution, which was found to effectively represent daily streamflow values. The distribution of daily flows was found to be modelled well in all seasons. However, the use of one distribution per day has the disadvantage of introducing a lot of parameters, which makes the model non-parsimonious

(Koutsoyiannis, 2016). If the user is not reliant on the generation of unobserved values, he/she might use the empirical instead of the theoretical kappa distribution for backtransformation instead. The use of the kappa distribution allows us to generate values that go beyond the range of observed values, which would not be the case if the empirical distribution was used. This ability of the generator to extrapolate extremes makes it suitable for applications where extreme events such as floods and droughts are of interest.

The generator can, on the one hand, be used to simulate streamflow at individual sites, and, on the other, to simulate jointly at multiple sites, which is not necessarily the case for other existing models. Its application to the example catchments, however, resulted in somewhat underestimated cross-correlations between stations. This underestimation can be explained by the fact that phase randomization preserves the cross-correlation in the normal domain but not necessarily in the domain of the original distribution. This cannot be overcome even if the simulation run which best reproduces these cross-correlations is extracted

from a large set of simulations. However, Stedinger and Taylor (1982) showed that estimators of the autocorrelation and cross-correlation of flows which do not match the historical sample estimates often provide more accurate estimates of the true but unknown correlations. Still, there are several potential avenues for improving the representation of cross-correlation. A first possibility would be the use of phase annealing (Hörning and Bárdossy, 2018). Phase annealing modifies the Fourier phases in an iterative way in order to optimize certain statistics, such as the cross-correlation function, and makes it possible to take

covariates into consideration for the generation of time series. However, using phase annealing increases the computational effort. A second possibility was presented by Keylock (2012) who only randomized the phases corresponding the wavelet coefficients lying above a certain threshold. He suggested to fix the large wavelet scales if one wanted to ensure that the low-frequency behavior between the observations and simulated series remains the same. This can indeed be a solution for retaining the cross-correlation between two series. However, it comes with the disadvantage that the temporal structure of the simulated

series is not very variable from the one of the observed series anymore. A third possibility is the introduction of functions correcting for the phase differences between two series as done by Nguyen et al. (2019) who applied this approach to correct for biases across multiple atmospheric variables derived from global circulation models. Another possibility for addressing the underestimation of cross-correlation would be the inflation of the cross-spectrum in the original domain in order to allow for a certain target cross-correlation after backtransformation. To do so, transformation approaches have been introduced, which

inflate the original process, which should after the backtransformation to the original domain result in a process with a target

distribution and correlation structure (Papalexiou, 2018; Tsoukalas et al., 2018b). An additional disadvantage of the method presented here (and of most other approaches presented in the literature) is that time irreversibility, which has been shown to be significant at a daily scale (Koutsoyiannis, 2019), is not explicitly modelled.

The streamflow generator was here used on observed streamflow time series. The input time series, however, do not nec-
essarily need to consist of observed values. One could also use the generator on streamflow simulated with a hydrological model. This extends its application to climate impact studies where a hydrological model is driven by meteorological time series generated with global and/or regional climate models. Alternatively, the representation of non-stationary conditions in the properties of the marginal distribution or the temporal dependence structure could also be achieved by adjusting the parameters of the marginal distribution or the frequency spectrum, respectively. Phase randomization simulation can potentially not only
accommodate changing climate conditions but also changes in land use or water extractions. The approach is not limited to the simulation of streamflow time series but extends to other hydro-meteorological variables such as precipitation, evapotranspiration, or snowmelt. This would require the test and identification of a suitable marginal distribution. In the case of intermittent processes, mixed-type marginal distributions would need to be used (Papalexiou, 2018). Distributions other than the kappa distribution can be used in PRSim by specifing a suitable (mixture) distribution. Spatio-temporal modelling of precipitation
fields, for example, may be performed using a technique based on phase randomization. However, it must be noted that due to the large number of zero observations (specifically with fine temporal resolution) the normal score transformation can become non-unique. In this case, additional efforts are needed to preserve the spatial structure of precipitation.

The stochastic streamflow generator presented here represents a flexible tool for streamflow simulation at individual or multiple sites. It can be used for various applications such as the design of hydropower reservoirs, the assessment of flood risk,
or the assessment of drought persistence and the estimation of the risk of multi-year droughts.

*Code availability.* The stochastic simulation procedure for a single site using the empirical, kappa, or any other distribution and some of the functions used to generate the validation plots are provided in the R-package PRSim. The stable version can be found in the CRAN repository https://cran.r-project.org/web/packages/PRSim/index.html, and the current development version is available at https://git.math. uzh.ch/reinhard.furrer/PRSim-devel.

*Data availability.* The observational discharge data was provided by the Federal Office for the Environment (FOEN), and can be ordered from http://www.bafu.admin.ch/wasser/13462/13494/15076/index.

*Author contributions.* AB and MB jointly developed the concept and methodology of the study. MB and RF set up the simulation approach. MB did the data analysis, produced the figures, and wrote the first draft of the manuscript. The manuscript was revised by RF and AB and edited by MB.

*Competing interests.* The authors do not have any competing interests

*Acknowledgements.* We acknowledge financial support provided by the FOEN (granted to MB via contract 15.0003.PJ/Q292-5096), the German Research Foundation (Deutsche Forschungsgemeinschaft, DFG) (granted to AB via project number Ba-1150/13-1 within the Space-Time Dynamics of Extreme Floods (SPATE) project), and the Swiss National Science Foundation (SNF) (granted to RF via project 175529). We thank the reviewers Ashish Sharma, Demetris Koutsoyiannis, and Simon Papalexiou for their constructive comments.

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
