# Peer review of "Technical note: Stochastic simulation of streamflow time series using phase randomization"

_Hydrology and Earth System Sciences, 2019_

## Short Comment (SC1) · 15 Apr 2019

This is only a very short suggestion about preserving ACF/CCF.

As the Authors recognize, "This underestimation can be explained by the fact that phase randomization preserves the cross-correlation in the normal domain but not necessarily in the domain of the original distribution."

Indeed, in my understanding the algorithm works like classical ARMA modelling procedures used in hydrological applications (replacing (stationary Gaussian) ARMA simulation with phase randomization).

[Figure]

Of course, a (strictly monotonic) marginal transformation preserves rank auto/cross-correlations but not the linear ACF/CCF of the target process (flow). An approach, which should not be computationally expensive (in principle), is to suitably inflate the spectrum of the "parent" Gaussian process. Since the ACF of a signal is the inverse Fourier transform of the power spectrum, such an inflated spectrum can result from suitably inflated ACF/CCF, which in turn can be obtained using the procedure described by Papalexiou (2018) and references therein. Such a technique only requires the solution of a simple double integral over a finite set of ACF/CCF values.

Therefore, in principle, it is possible to define the ACF/CCF terms (and thus the spectrum terms) of a parent Gaussian process yielding the required ACF/CCF of the target process, avoiding brute-force procedures. Compared with Papalexiou's procedure, in the present case, you only need an additional transformation (from Gauss inflated (empirical) ACF to Gauss inflated (empirical) spectrum) in order to use phase randomization instead of VAR(p) models (applied by Papalexiou), but I think that this is a matter of technicalities.

Sincerely,

Francesco

PS: the Authors can also be interested in Koutsoyiannis (2019), discussing the problem of temporal asymmetry, which is related to more realistic stream flow simulations and is in line with recent Andras' works on the topic, I think.

References

Koutsoyiannis D. (2019) Time's arrow in stochastic characterization and simulation of atmospheric and hydrological processes, Hydrological Sciences Journal, doi:10.1080/02626667.2019.1600700.

Papalexiou, S. M. (2018). Unified theory for stochastic modelling of hydroclimatic processes: Preserving marginal distributions, correlation structures, and intermittency.

[Figure]

Advances in water resources, 115, 234-252.

---

## Short Comment (SC2) · 19 Apr 2019

I enjoyed reading this paper, which aims to address a challenging and critical task for many water resources studies, that of synthetic data generation. I also appreciated the availability of the PRSim R package. This is an additional short comment continuing the interesting comment posed by F. Serinaldi regarding the preservation of the ACF/CCF. As the Authors recognize, and F. Serinaldi mentions in his comment, the ACF/CCF in the Gaussian domain and the ACF/CCF in the actual domain (i.e., that of the distribution function), typically differ. In my view, this is mainly due to two reasons:

[Figure]

1. The final process (the one obtained after the use of the quantile function; i.e., the inverse cumulative distribution function; ICDF) results from the mapping of a Gaussian one (a concept related with the so-called Nataf's joint distribution model (Mardia, 1970; Nataf, 1962) – see below)

2. The use of Pearson's correlation coefficient to express the dependence structure (ACF/CCF) of the process.

Within hydrological domain, beyond the work of Papalexiou (2018) (also mentioned by F. Serinaldi), this mapping procedure has been also employed for the simulation of non-Gaussian univariate and multivariate processes in the works of Tsoukalas et al. (2019, 2018a, 2018b, 2017) and Tsoukalas (2018; for a Thesis-length treatment on the subject) which adopt the term "Nataf-based processes", and also discuss similarities with other approaches in hydrology and beyond (see also the work of Serinaldi and Lombardo (2017) for univariate binary processes). For instance, the work of Liu and Der Kiureghian (1986) and Lebrun and Dutfoy (2009), that regard the Nataf's joint distribution model per se – which in my understanding regards the core idea of the above mapping procedure (initially proposed for correlated random variables, and not processes).

Interestingly, and regarding processes, the concept of Nataf's joint distribution can be traced back in the early work of Grigoriu (1984) and later in sequel works, who used it to establish non-Gaussian stochastic process, the so-called "translation processes". Similar models for non-Gaussian stochastic processes have been proposed by Cario and Nelson (1996) and Biller and Nelson (2003), the so-called "To-Anything" models. More specifically, this type of models use (low-order) AR models to simulate an auxiliary (appropriately "inflated/adjusted") Gaussian process (Gp), which after its mapping (through the ICDF) to the actual domain, results into a process with the target distribution and correlation structure. Actually, it is noted, that any linear stochastic model (e.g., AR, MA, ARMA or other) can be used within such simulation schemes to simulate

the auxiliary Gp (Tsoukalas et al., 2018b).

Regarding the preservation of the ACF/CCF from such methods (i.e., relying on this mapping procedure), it is recalled that the Pearson's correlation is a linear measure of dependence that uses first cross-product moments of the process, and is not invariant under non-linear monotonic transformations (such as those imposed by the ICDF). Please, also refer to section 4 in Tsoukalas et al. (2018c), and section 3.2.3 in Tsoukalas et al. (2018b), who highlight a delicate point related with the above mapping procedure and the use of alternative rank-based dependence measures (i.e., Kendall's tau and Spearman's rho). Further information can be found in the book of Embrechts et al. (1999), while a discussion related with this property, in the context of similarly-constructed stochastic processes for hydrological time series generation, is given in Tsoukalas et al. (2018a, 2018b).

As noted in F. Serinaldi's comment, and also discussed by others (e.g., Koutsoyiannis, 2016, 2000; Papoulis, 1991 pp. 118) the power spectrum and the ACF (or CCF) of a process are interrelated quantities. Hence, it is reasonably to expect that a spectrum-based simulation method (such as the one proposed by the Authors) will inherit the properties of Pearson's correlation coefficient. To elaborate, let $Z_t$ be a univariate stationary standard (with mean zero and unit variance) Gaussian process with auto-correlation $\widetilde{\rho}_\tau = \text{Corr}(X_t, X_{t+\tau})$, where $\tau$ denotes the time lag, and $X_t$ be a process obtained by the mapping operator $X_t = F^{-1}(\Phi(Z_t))$, where $F^{-1}()$ denotes the ICDF of the target distribution (with finite variance) and $\Phi()$ denotes the Gaussian cumulative distribution function. It can be shown that the autocorrelation $\rho_\tau = \text{Corr}(X_t, X_{t+\tau})$ of the final process is related to the Gaussian one by (see the abovementioned papers, and references therein),

$$\rho_\tau = \frac{\int_{-\infty}^{\infty} \int_{-\infty}^{\infty} F_X^{-1}(\Phi(z_t)) F_X^{-1}(\Phi(z_{t+\tau})) \ \varphi_2(z_t, z_{t+\tau}, \ \widetilde{\rho}_\tau) \, dz_t dz_{t+\tau} - (E[X])^2}{\text{Var}[X]} 1 \quad (1)$$

where $\varphi_2(z_t, z_{t+\tau}, \ \widetilde{\rho}_\tau)$ is the bivariate standard normal probability density function,

which contains $\widetilde{\rho}_\tau$.

Furthermore let $\gamma_\tau = \mathrm{Cov}\left(X_t, X_{t+\tau}\right) = \rho_t\gamma_0$ denote the autocovariance of the process, where $\gamma_0$ stands for its variance.

Let also recall that the power spectrum and the autocorrelation function of a process are related by (e.g., Koutsoyiannis, 2016, 2000; Papoulis, 1991 pp. 118),

$$S_{\gamma(\omega)} = 2\gamma_0 + 4\sum_{\tau=1}^{\infty}\gamma_\tau\cos\left(2\pi\tau\omega\right) = 2\sum_{\tau=-\infty}^{\infty}\gamma_\tau\cos\left(2\pi\tau\omega\right), \quad \omega \in [0,\ 1/2]\ 2 \quad (2)$$

Therefore, by using $\gamma_\tau = \rho_t\gamma_0$ and substituting Eq. (1) into Eq. (2), it is shown that the power spectrum of the process $X_t$ is related to the autocorrelation $\widetilde{\rho}_\tau$ of $Z_t$, and hence on its power spectrum $S_{\widetilde{\gamma}(\omega)} = 2\sum_{\tau=-\infty}^{\infty}\widetilde{\gamma}_t\cos\left(2\pi\tau\omega\right)$, where $\widetilde{\gamma}_t = \widetilde{\rho}_\tau$ (since the Gp has unit variance).

For completeness, it is mentioned that the autocovariance $\gamma_\tau$ can be obtained from a known power spectrum $S_{\gamma(\omega)}$ by,

$$\gamma_\tau = \int_0^{1/2} S_{\gamma(\omega)}\cos\left(2\pi\tau\omega\right)\mathrm{d}\omega\ , \quad j = 0, 1, 2, \ldots .3 \quad (3)$$

An example of a spectrum-based method that uses this mapping procedure (i.e., through the ICDF) in combination with a suitably inflated spectrum for the auxiliary (or parent) Gaussian process is given by Deodatis and Micaletti (2001). This method avoids the underestimation of correlation coefficients and manages to preserve the distribution function of the process.

Just a few quick comments:

1. Section 3.1 (step 1): Since the authors employ the Kappa distribution (a generalization of the GEV distribution; which is bounded from below or above, depending on the parameters) to model the historical data, it could be insightful to mention that under certain parameter combinations, this distribution may lead to infinite moments. This can be a delicate issue, since if the fitted distribution exhibits infinite variance then the Pearson's correlation cannot be defined (the denominator contains the variance), and thus the proposed model (as well as many other models) cannot be used. This situation is discussed in section 3.4 of Tsoukalas et al. (2018b; and references therein), where it is advocated (based on empirical, as well as theoretical reasoning) that physical processes are characterized by finite variance (Koutsoyiannis, 2016).

Particularly, if $X$ is a Kappa-distributed random variable, and $\mu_r = E\left[X^r\right]$ denotes the $r^{th}$ raw moment, as discussed in Hosking (1994), and elsewhere, the existence of the $r^{th}$ depends on the values of $h$ and $k$. Specifically, the moments exist:

$$\text{for all } r \quad \text{if} \quad h \geq 0 \text{ and } k \geq 0$$

$$\text{for } r < -1/hk \quad \text{if} \quad h < 0 \text{ and } k \geq 0, \text{ and}$$

$$\text{for } r < -1/k \text{ if } k < 0$$

It is also interesting to mention that Hosking (1994) notes that the first four moments cannot uniquely determine the parameters of the distribution, since some combinations of moments (expressed by skewness and kurtosis coefficients) correspond to different pairs of $h$ and $k$.

1. Section 3.1 (step 1): Can you please provide more details on the employed fitting method. The fitting was performed using classical product-moments, L-moments, maximum likelihood, or another method?

2. Section 3.1 (Introduction and step 2): I believe that it would be useful to mention that the recent literature contains alternative models, such as the Stochastic Periodic Autoregressive To Anything (SPARTA) model of Tsoukalas et al. (2018a, 2017), that do not employ de-seasonalization techniques and are able to simulate cyclostationary processes (univariate and multivariate), accounting for many of its facets such as, seasonally varying marginal distributions and correlations.

3. Section 3.1 (step 6): The Authors mention that: "Negative simulated values are replaced by 0, which corresponds to the lower boundary of the Kappa distribution". As far as I am aware the left support of Kappa distribution is not necessarily zero (e.g., when $k = 0$ and $h \leq 0$, then the supports of the distribution are, $-\infty < x < \infty$; see Hosking (1994)). In any case, the generation of negative values can be eliminated by using a distribution function defined in the positive real line. Particularly, I would suggest the investigation/use of the Generalized Gamma and Burr type-XII distributions, which are more parsimonious (they entail three parameters) and were found adequate for modelling of hydrometeorological variables; particularly rainfall (e.g., Papalexiou and Koutsoyiannis, 2016). Examples of their use within the context of stochastic modelling can be found the work Papalexiou (2018), as well as in Tsoukalas et al. (2019, 2018b) and Tsoukalas (2018).

Regards,

Ioannis Tsoukalas

**References**

Biller, B., Nelson, B.L., 2003. Modeling and generating multivariate time-series input processes using a vector autoregressive technique. ACM Trans. Model. Comput. Simul. 13, 211–237. https://doi.org/10.1145/937332.937333

Cario, M.C., Nelson, B.L., 1996. Autoregressive to anything: Time-series input processes for simulation. Oper. Res. Lett. 19, 51–58. https://doi.org/10.1016/0167-6377(96)00017-X

Deodatis, G., Micaletti, R.C., 2001. Simulation of highly skewed non-Gaussian stochastic processes. J. Eng. Mech. 127, 1284–1295.

Embrechts, P., McNeil, A.J., Straumann, D., 1999. Correlation and Dependence in Risk Management: Properties and Pitfalls, in: Dempster, M.A.H. (Ed.), Risk Management. Cambridge University Press, Cambridge, pp. 176–223. https://doi.org/10.1017/CBO9780511615337.008

Grigoriu, M., 1984. Crossings of Non-Gaussian Translation Processes. J. Eng. Mech. 110, 610–620. https://doi.org/10.1061/(ASCE)0733-9399(1984)110:4(610)

Hosking, J.R.M., 1994. The four-parameter kappa distribution. IBM J. Res. Dev. 38, 251–258. https://doi.org/10.1147/rd.383.0251

Koutsoyiannis, D., 2016. Generic and parsimonious stochastic modelling for hydrology and beyond. Hydrol. Sci. J. 61, 225–244. https://doi.org/10.1080/02626667.2015.1016950

Koutsoyiannis, D., 2000. A generalized mathematical framework for stochastic simulation and forecast of hydrologic time series. Water Resour. Res. 36, 1519–1533. https://doi.org/10.1029/2000WR900044

Lebrun, R., Dutfoy, A., 2009. An innovating analysis of the Nataf transformation from the copula viewpoint. Probabilistic Eng. Mech. 24, 312–320. https://doi.org/10.1016/j.probengmech.2008.08.001

Liu, P.L., Der Kiureghian, A., 1986. Multivariate distribution models with prescribed marginals and covariances. Probabilistic Eng. Mech. 1, 105–112. https://doi.org/10.1016/0266-8920(86)90033-0

Mardia, K. V, 1970. A Translation Family of Bivariate Distributions and Frechet's Bounds. Sankhya Indian J. Stat. Ser. A 32, 119–122.

Nataf, A., 1962. Statistique mathematique-determination des distributions de probabilites dont les marges sont donnees. C. R. Acad. Sci. Paris 255, 42–43.

Papalexiou, S.M., 2018. Unified theory for stochastic modelling of hydroclimatic processes: Preserving marginal distributions, correlation structures, and intermittency. Adv. Water Resour. https://doi.org/10.1016/j.advwatres.2018.02.013

Papalexiou, S.M., Koutsoyiannis, D., 2016. A global survey on the seasonal variation of the marginal distribution of daily precipitation. Adv. Water Resour. 94, 131–145. https://doi.org/10.1016/j.advwatres.2016.05.005

Papoulis, A., 1991. Probability, Random Variables, and Stochastic Processes, Third edit. ed. McGraw-Hill Series in Electrical Engineering. New York City, New York, USA: McGraw-Hill.

Serinaldi, F., Lombardo, F., 2017. BetaBit: A fast generator of autocorrelated binary processes for geophysical research. EPL (Europhysics Lett. 118, 30007. https://doi.org/10.1209/0295-5075/118/30007

Tsoukalas, I., 2018. Modelling and simulation of non-Gaussian stochastic processes for optimization of water-systems under uncertainty. PhD Thesis, Department of Water Resources and Environmental Engineering, National Technical University of Athens (Defence date: 20 December 2018).

[Figure]

Tsoukalas, I., Efstratiadis, A., Makropoulos, C., 2019. Building a puzzle to solve a riddle: A multi-scale disaggregation approach for multivariate stochastic processes with any marginal distribution and correlation structure. J. Hydrol (in review).

Tsoukalas, I., Efstratiadis, A., Makropoulos, C., 2018a. Stochastic Periodic Autoregressive to Anything (SPARTA): Modeling and simulation of cyclostationary processes with arbitrary marginal distributions. Water Resour. Res. 54, 161–185. https://doi.org/10.1002/2017WR021394

Tsoukalas, I., Efstratiadis, A., Makropoulos, C., 2017. Stochastic simulation of periodic processes with arbitrary marginal distributions, in: 15th International Conference on Environmental Science and Technology. CEST 2017. Rhodes, Greece.

Tsoukalas, I., Makropoulos, C., Koutsoyiannis, D., 2018b. Simulation of stochastic processes exhibiting any-range dependence and arbitrary marginal distributions. Water Resour. Res. https://doi.org/10.1029/2017WR022462

Tsoukalas, I., Papalexiou, S., Efstratiadis, A., Makropoulos, C., 2018c. A Cautionary Note on the Reproduction of Dependencies through Linear Stochastic Models with Non-Gaussian White Noise. Water 10, 771. https://doi.org/10.3390/w10060771

---

## Short Comment (SC3) · 19 Apr 2019

Dear Ioannis,

I could not say better!... and in fact, I did not :-) I have to say that I was not aware of some the references you mention (papers on PEM and JEM journals). I hope that the Authors will benefit from your review. From mi side, I did. So, thanks!

Cheers

F

---

## Short Comment (SC4) · 28 Apr 2019

Dear Authors,

This is an interesting paper on the simulation of daily streamflow through power-spectrum-based Gaussian implicit scheme and the Kappa distribution. The results indicate that the proposed scheme can very well preserve the marginal distribution and certain aspects of the frequency-domain second-order dependence structure.

The simulation algorithm is applied through the phase spectrum randomization (instead of the white noises used in autoregressive moving-average stochastic models), while it preserves the expected frequency spectrum (and thus, the expected second-order dependence structure, e.g. autocovariance function) of each process as well as the cross-correlation between two processes, whereas the Kappa marginal distribution is preserved through the Gaussian-implicit method (i.e. by transforming the target distribution to the Gaussian one and at the end, use the inverse transformation to go back to the target one).

Please see below some short/long comments to initiate a discussion which I hope they can contribute to the proposed methodology and further highlight it.

1) In the main text, the Authors mention several existing streamflow generation approaches that have focused on the frequency domain (instead on the more traditional time-domain), with the earliest cited work of Theiler et al. (1992, see ref. in the text).

As mentioned by the Authors the Gaussian-implicit (also mentioned as copula) can be implemented not only through the autocorrelation function (lag-domain) but also to the power-spectrum (frequency-domain), which are theoretically linked through the Fourier transform. Another application could be through the variance (scale-domain) of the process, else known as climacogram (i.e. the variance of the averaged process vs. time scale of averaging, see also the $6^{th}$ comment). These three stochastic tools are linked theoretically and if one is known the rest can be easily obtained (Koutsoyiannis, 2016):

$$c(h) = \int_0^\infty s(w) \cos(2\pi wh) \, dw \leftrightarrow s(w) = 4 \int_0^\infty c(h) \cos(2\pi wh) \, dh \leftrightarrow$$

$$s(w) = -2 \int_0^\infty (2\pi wk)^2 \gamma(k) \cos(2\pi wk) \, dk \leftrightarrow \gamma(k) = \int_0^\infty s(w) \frac{sin^2(\pi wk)}{(\pi wk)^2} dw$$

where $c$, $s$ and $\gamma$ are the autocovariance function, the power-spectrum and the climacogram, and $h$, $w$ and $k$ are the continuous time-lag, time-frequency and time-scale (i.e. the three major time domains for statistical analysis), respectively.

However, although the above three stochastic tools can be analytically and equivalently-easy obtained from one another, *they do not exhibit the same statistical robustness in parameter estimation from data*. Like for example the central moments are considered less accurate than the L-moments (i.e. with a lower statistical bias), the identification of the second-order dependence structure is more accurately estimated by the climacogram (for comparison of several short and long-term processes and the reasons why the performance of climacogram in stochastic modelling should be preferred see Dimitriadis and Koutsoyiannis, 2015a).

In other words, one could estimate the model of the second-order dependence structure from the empirical climacogram and then, find the same model expressed through the power-spectrum from the above equations (or even directly from data through the climacospectrum; Koutsoyiannis, 2016, 2019a). In this way, the selected model is expected to be closer to the theoretical model.

2) Another issue maybe worth discussing is the *statistical bias* of the above estimators of the autocovariance, power-spectrum and climacogram. In case of a long-term process the autocovariance is highly biased (e.g., Dimitriadis and Koutsoyiannis, 2015a) and therefore, the expectation of the Fourier estimator (linked to the power-spectrum) will be also highly biased leading to underestimation (as the Authors observed in the cross-correlations among stations). I would suggest either to add this to the Discussion for future research or to fit a model to the observed power-spectrum (or better to the climacogram as discussed above), then estimate the bias of the power-spectrum, and then perform the simulation based on the latter (as suggested by a general framework for Stochastics in Koutsoyiannis, 2000).

3) Also, by fitting a model to the power-spectrum the Authors will be able to estimate the *Hurst parameter*, which characterizes the long-term behaviour (e.g. Koutsoyiannis, 2008). Such a comment about this Hurst parameter could be added to the text, since the Authors specifically focus on processes with long-term behaviour, which is also called Hurst-Kolmogorov behaviour (Koutsoyiannis, 2016). Interestingly this behaviour has been identified through long-term stochastic similarities in 13 processes in global and local scales in the PhD thesis of Dimitriadis (2017, see also Koutsoyiannis et al., 2018), where it is also proposed an integrated stochastic view among various geophysical and hydrometeorological processes which may differ in their physical properties but not in their stochastic behaviour). Also, for the long-term or HK processes see an additional recent literature review of O'Connell et al. (2016).

4) Concerning the concept of de-seasonalization the Authors may find worth mentioning the term of *homogenization* (e.g. Dimitriadis and Koutsoyiannis, 2018, sect. 4.2), which is actually the same concept as de-seasonalization but also includes cases of double periodicities (i.e. diurnal and seasonal) as well any other implicit transformation, i.e. transform the process to a desired

distribution for simulation G($x_{i,j}$), where $i$ and $j$ are indices for the seasonal and diurnal processes, and then go back to the original distribution F($x_{i,j}$) using the inverse transformation F(G$^{-1}$($x_{i,j}$)). These transformations are usually based on a target marginal distribution (e.g. maximized-entropy distributions in case the distribution function is unknown or other non-Gaussian distributions), and because this method uses the inverse transformation of G($x_{i,j}$) it is called implicit algorithm to be separated from the explicit algorithms (Koutsoyiannis and Dimitriadis, 2016) which preserve simultaneously (not in an indirect manner) an arbitrary second-order dependence structure and a desired marginal distribution through its marginal moments (for such algorithms see the recent work of Dimitriadis and Koutsoyiannis, 2018).

5) Maybe could be helpful for the Readers to add some *Tables* with the fitted parameters of the Kappa distribution for each season and each river. It has to be noted though that the Kappa distribution with $\xi > 0$ seem to fit the streamflow for $x > \xi$ and thus, does not include the zero values. Maybe a bivariate two-state approach (streamflow – no-streamflow, as for example in Li et al., 2012 for precipitation, where they also use the Kappa distribution). In case the Authors choose a one-state approach then they may find useful the discussion and justification for the threshold parameter to daily precipitation in Dimitriadis and Koutsoyiannis (2018, sect. 4.3), where a one-state approach is applied for the Pareto-Burr-Feller (PBF) distribution with a threshold parameter $h$.

6) The Authors may find useful for extending their literature review the work of Cugar and Kavanagh (1968), who seem to be the first ones in literature to apply the Gaussian-implicit scheme at the frequency-domain. Also, there seems to be an earlier work of Hoeffding (1940) that has a more general expression for implicit algorithms (please see further relevant literature in Dimitriadis and Koutsoyiannis, Appendix D).

Also, maybe is worth noticing that the *Gaussian-implicit through the scale-domain* (climacogram) for the simulation of a single timeseries has been discussed in Dimitriadis (2017, sect. 3.3.3, and references therein) and Dimitriadis and Koutsoyiannis (2018, Appendix D), and earlier illustrated for the maximized entropy distribution function (in case the marginal distribution is unknown) in Dimitriadis and Koutsoyiannis (2015b, see also Koutsoyianni et al., 2008), as well as for the extended Pareto marginal distribution (through a simple application) of a double periodic hourly wind process in Deligiannis et al. (2016, see Fig. 8a with the transformation of the empirical climacogram following an extended Pareto distribution -red line- to the climacogram following a Gaussian distribution -black line-):

[Figure]

Finally, the Authors may find interesting recent work of Iliopoulou et al. (2018) on seasonal precipitation extremes and Iliopoulou et al. (2018) on seasonal streamflow correlations.

7) A final comment worth discussing is that the Gaussian-implicit methodology may sometimes severe *overestimate the variability/uncertainty* of highly skewed distributions (such as in daily river discharge as discussed by the Authors, see also Koutsoyiannis, 2019b). When transforming the generated Gaussian process to the target distribution through the inverse function, the expectation of the 2nd order dependence structure (e.g. mean value of the power spectrum estimator) maybe well preserved but its variability is likely that it is not adequately preserved. This is a basic limitation of the Gaussian-implicit schemes and has been highlighted in Dimitriadis and Koutsoyiannis (2018). The reason is that the dependence structure depends on the high-order moments by definition and thus, a transformation could not simultaneously alter both the marginal and the dependence structure. This may be tackled by explicit methodologies of stochastic simulation (Dimitriadis and Koutsoyiannis, 2018; also see an illustrative example in Appendix D) but the latter cannot preserve in an theoretically exact way the marginal distribution since it is based on the preservation of moments rather than the marginal distribution function itself (like in implicit schemes). A fancy solution could be the use of higher-order copula (as also mentioned in Dimitriadis and Koutsoyiannis, 2018, Appendix D).

As a final comment I would like to congratulate their Authors for their work, for their ideas discussed, and for not being carried away using vague expressions in their text like *to anything*, *unified theory*, *ill transformation* etc. As always in Science every algorithm has merits and demerits and only through open discussion, we may invent new algorithms for engineering practices that keep the merits and overcome the demerits.

Sincerely,

Panayiotis Dimitriadis

*References*

Gugar, V.G. and R.J. Kavanagh, Generation of random signals with specified probability density functions and power density spectra, *IEEE-AC*, 13 :716–719, 1968.

Deligiannis, I, P. Dimitriadis, O. Daskalou, Y. Dimakos, and D. Koutsoyiannis, Global investigation of double periodicity of hourly wind speed for stochastic simulation; application in Greece, *Energy Procedia*, 97, 278–285, 2016.

Dimitriadis, P., Hurst-Kolmogorov dynamics in hydrometeorological processes and in the microscale of turbulence, PhD thesis, 167 pages, *National Technical University of Athens*, Athens, doi: 10.13140/RG.2.2.34652.697682017.

Dimitriadis, P., and D. Koutsoyiannis, Climacogram versus autocovariance and power spectrum in stochastic modelling for Markovian and Hurst–Kolmogorov processes, *Stochastic Environmental Research & Risk Assessment*, 29 (6), 1649–1669, 2015a.

Dimitriadis, P., and D. Koutsoyiannis, Application of stochastic methods to double cyclostationary processes for hourly wind speed simulation, *Energy Procedia*, 76, 406–411, 2015b.

Dimitriadis, P., and D. Koutsoyiannis, Stochastic synthesis approximating any process dependence and distribution, *Stochastic Environmental Research & Risk Assessment*, 32 (6), 1493–1515, 2018.

Hoeffding, W., Scale-invariant correlation theory, in N. I. Fisher and P. K. Sen (Eds.), The Collected Works of Wassily Hoeffding, pp. 57–107, *New York: Springer-Verlag*, 1940.

Iliopoulou, T., D. Koutsoyiannis, and A. Montanari, Characterizing and modeling seasonality in extreme rainfall, *Water Resources Research*, 54 (9), 6242–6258, 2018.

Iliopoulou, T., C. Aguilar , B. Arheimer, M. Bermúdez, N. Bezak, A. Ficchi, D. Koutsoyiannis, J. Parajka, M. J. Polo, G. Thirel, and A. Montanari, A large sample analysis of European rivers on seasonal river flow correlation and its physical drivers, *Hydrology and Earth System Sciences*, 23, 73–91, 2019.

Koutsoyiannis, D., A generalized mathematical framework for stochastic simulation and forecast of hydrologic time series, *Water Resources Research*, 36 (6), 1519–1533, 2000.

Koutsoyiannis, D., Hurst-Kolmogorov dynamics and uncertainty, *Journal of the American Water Resources Association*, 47 (3), 481–495, 2011.

Koutsoyiannis, D., Generic and parsimonious stochastic modelling for hydrology and beyond, *Hydrological Sciences Journal*, 61 (2), 225–244, 2016.

Koutsoyiannis, D., Knowable moments for high-order stochastic characterization and modelling of hydrological processes, *Hydrological Sciences Journal*, 64 (1), 19–33, 2019a.

Koutsoyiannis, D., Time's arrow in stochastic characterization and simulation of atmospheric and hydrological processes, *Hydrological Sciences Journal*, 2019b.

Koutsoyiannis, D., and P. Dimitriadis, From time series to stochastics: A theoretical framework with applications on time scales spanning from microseconds to megayears, *Orlob Second International Symposium on Theoretical Hydrology*, Davis, California, USA, doi:10.13140/RG.2.2.14082.89284, University California Davis, 2016.

Koutsoyiannis, D., H. Yao, and A. Georgakakos, Medium-range flow prediction for the Nile: a comparison of stochastic and deterministic methods, *Hydrological Sciences Journal*, 53 (1), 142–164, 2008.

Koutsoyiannis, D., P. Dimitriadis, F. Lombardo, and S. Stevens, From fractals to stochastics: Seeking theoretical consistency in analysis of geophysical data, *Advances in Nonlinear Geosciences*, edited by A.A. Tsonis, 237–278, doi:10.1007/978-3-319-58895-7_14, Springer, 2018.

Li, C., V.P. Singh, and A.K. Mishra, Simulation of the entire range of daily precipitation using a hybrid probability distribution, *Water Resources Research*, 48, W03521, 2012.

O'Connell, PE, D. Koutsoyiannis, H. F. Lins, Y. Markonis, A. Montanari, and T.A. Cohn, The scientific legacy of Harold Edwin Hurst (1880 – 1978), *Hydrological Sciences Journal*, 61 (9), 1571–1590, 2016.

---

## Referee Comment (RC1) · Ashish Sharma (Referee) · 29 Apr 2019

My congratulations to the authors on this excellent paper. Very glad to see a clever adopted to frequency domain alternatives in formulating a stochastic streamflow generator. My comments below are aimed to enhance the presentation and I am in support of publication once these have been addressed. Comments are:

line 2/9 - The authors are missing the works by Keylock (10.1029/2012WR011923). This work performed resampling to an existing time series using phase randomisation in the frequency domain. If I remember correctly, it had some nice inclusion of ICA to tackle the multivariate issue, and wavelets to get around nonstationarity in the data

that cannot be handled using a fourier transformation alone. I think they need to read those papers (I am familiar with the above one bnut there may be more since) and acknowledge them here, and also try and show how their work distinguishes itself from the above paper.

line 3/21: I think the work by Mehrotra (10.1029/2005JD006637) should be acknowledged here as it represents essentially something analogous to a ARMAX type of a model even though it is cast as a stochastic downscaling approach. A mention should be made on the ability to preserve low frequency variability, which I believe the proposed approach will be able to address as well.

Line 3/35: Even though it relates to the problem of correcting systematic biases, given the use of phase transformation (not randomisation), the approaches of Nguyen should perhaps be acknowledged for completeness. The rationale behind these approaches and the one here has a lot in common. (10.1007/s00382-018-4191-6, 10.1016/j.jhydrol.2016.04.018).

line 5/21: The authors may want to look through the details of (10.1007/s00382-018-4191-6, 10.1016/j.jhydrol.2016.04.018) as they performed another level of pre-processing - they fit a Thomas Feiring type model to the monthly data and after that structure was removed, the Fourier transformation was performed. This was done after trying with the steps referred to above, as it was found to exhibit clear advantages.

line 6/21: Setting negatives to zero is not a clean option. Please refer to the Keylock paper above again on how they restricted their approach to resampling to avoid having to set negatives to zero. line 11/10: Underestimation of cross-correlations is I think addressed well in (10.1007/s00382-018-4191-6). The trick that is used is to not randomly generate phases for all variables, but for a "key" variable (say biggest streamflow mean location). And then maintain the phase difference between alternate sites. The phase difference in space helps capture the cross-dependence attributes.

Lastly, I feel not addressing the issue of non-stationarity in a stochastic generation

paper under our present climate should be discouraged. The issue of nonstationarity can be addressed in the sense of a discussion by thinking of adding an exogenous predictor variable set in the formulation, which can impart the changes needed. Some discussion to that effect would be good to include in the paper before it is published.

---

## Referee Comment (RC2) · Demetris Koutsoyiannis (Referee) · 12 May 2019

**Review report on "Stochastic simulation of streamflow time series using phase randomization"**

by Demetris Koutsoyiannis

| | |
|---|---|
| Journal: | Hydrol. Earth Syst. Sci. Discuss. |
| Journal's Ref.: | HESS-2019-142 |
| Title: | Technical note: Stochastic simulation of streamflow time series using phase randomization |
| Authors: | Manuela I. Brunner, András Bárdossy, and Reinhard Furrer (Switzerland, Germany) |
| Reviewer's Ref.: | DK-JR-345 |
| Date: | 2019-05-12 |
| Recommendation: Major revision | |

Reviewer's assertion: It is my opinion that a shift from anonymous to eponymous (signed) reviewing would help the scientific community to be more cooperative, democratic, equitable, ethical, productive and responsible. Therefore, it is my choice (and aesthetic attitude) to sign my reviews.

1. The Technical Note by Brunner et al. (2019) implements a useful idea for easy stochastic simulation of daily streamflow, based on spectral representation and phase randomization. The method has several limitations (see below) but it is practical and useful, and it certainly deserves publication. I believe several issues can be improved before final publication and therefore I am providing some suggestions. I also appreciate the commentaries by Francesco Serinaldi, Ioannis Tsoukalas and Panayiotis Dimitriadis, who provided a lot of information to the authors. I think this information is useful to optimize their Note and also to put it in the context of modern and older literature, some of which is missing in the literature review. I believe that not everything suggested in the commentaries needs to be addressed, as this would change the orientation of the Note. However, with several changes in the formulations and a few expansions, rather than additional analyses, the Note could be improved. My own suggestions, which I list in the following points, fall in two categories: (a) recognition of the limitations of the method and (b) improvements in formulations, phraseology and terminology.

2. A first limitation, which in the current version is not stated clearly, is the severe dependence of the method on the sample size of observations. The synthetic series has the same length as the observed series. The authors properly recognize the importance of respecting long-range dependence (LRD) in simulation. However, to study its effect in hydrosystems we need synthetic series much longer than the observed. The use of ensembles of small-length time series may not be equivalent

with using a long time series as each member of the ensemble is independent from the others.

3. A second limitation is the absence of a model for time dependence. While the authors correctly adopt a model for the marginal distribution (e.g. they state "*Using the empirical distribution instead of the Kappa distribution would prevent us from obtaining values that go beyond the range of observed data…*") their method misses to do so for the dependence structure. The empirical autocorrelogram and periodogram are affected by significant bias and huge noise (see references provided by Panayiotis Dimitriadis) and if we do not use a model, then we reproduce a particular random realization, in terms of autocorrelogram and periodogram, in all our simulated series. I believe authors' statement "*The periodogram, the empirical counterpart of the power spectrum, shows high values at those frequencies which correspond to strong periodic components*" is only partly true and perhaps misleading. The periodogram could be regarded a realization of a stochastic process per se (on the frequency domain) and its peaks do not necessarily reflect a real peak in the "true" power spectrum. The same thing happens with the autocorrelogram. For example, the ups and downs in the empirical autocorrelograms in Fig. 5 may well be sampling artefacts, which we do not need to reproduce—but the method does reproduce them.[1]

4. A third limitation is the lack of parsimony of the entire methodology. From the statement "*We fit a separate distribution for each day to take into account seasonal differences in the distribution of daily streamflow values*" one can imagine that the overall method encompasses lots of parameters. Apparently, it is nowadays easy to do calculations with lots of parameters but, in my view, stochastics goes beyond calculations and algorithms. Parsimony in stochastic modelling is always important (see Koutsoyiannis 2016).

5. A final limitation for the particular time scale of modelling, i.e. daily, is the lack of explicit modelling of time irreversibility (an issue also mentioned in the comment by Francesco Serinaldi). This would not be an issue if the time scale was monthly or longer, but I suspect that it is for the daily scale (see Koutsoyiannis 2019 and also Müller et al. 2017). I clarify here that I do not suggest changing the method to overcome the limitations (e.g. to become more parsimonious or to take irreversibility into account). Rather, I just recommend stating them in a clear and explicit manner.

6. Now coming to the second category of my suggestions, I would recommend avoiding the name *kappa distribution* for the chosen distribution. It is true that in hydrological literature this name is in common use, but if we wish to facilitate communication with other disciplines, we should be aware that the name *kappa*
* * *
[1] If the authors have difficulty to accept my comment, I would suggest doing an experiment with a particular (smooth) autocorrelation function and see the ups and downs in the produced autocorrelogram and periodogram of a single realization.

*distribution* has another meaning in statistical thermodynamics—namely it is used to describe Cauchy-type (or Student-like) distributions in motion of particles (e.g. Olbert, 1968; Livadiotis and McComas 2013). The specific distribution used in the Note (which I do not think is a generalization of GEV as suggested by Ioannis Tsoukalas), is commonly (in most disciplines) referred to as the Dagum distribution—see https://en.wikipedia.org/wiki/Dagum_distribution. In addition, in terms of sign conventions in eqn. (4), I would suggest changing the signs of $k$ and $h$ and replacing the two minus signs in front of them with plus signs. This will make the expression more convenient and intuitive, and also complying to the standard notation used in other disciplines (e.g. as seen in the above web site).

7. The phrase "*Stochastically generated time series mimic the characteristics of observed data and represent sets of plausible but **as yet unobserved** streamflow sequences*" (my emphasis) may distort the meaning of what stochastic simulation is. It is not a matter of something that is "yet unobserved" but expected to be observed in the future. It is a matter of producing artificial "realizations" from the stochastic model. A model, stochastic or otherwise, is not identical to the real world.

8. The term "deseasonalization" needs to be used with care and clarification; otherwise it may mislead people to think that, by techniques like that used in the Note, we can get rid of seasonality. This, however, is quite difficult—if ever possible. With transformations of the time series, either linear (as in standardizing by mean and variance of each period) or nonlinear (as in fitting a separate distribution for each period, like what is done in this Note), we can only remove the seasonal effect on the marginal distribution, not that of the joint distribution of a cyclostationary stochastic process. (For example, differences in autocorrelation coefficients in different seasons are not removed by techniques such as the above mentioned). Therefore I suggest replacing "deseasonalization" with "deseasonalization of the marginal distribution."

9. The notion of "nonparametric" techniques referred to in the literature review is, in my opinion, problematic when we deal with stochastic processes with time dependence. As opposite to iid statistics, in which the first "i" (independent) is taken for granted, in stochastics there cannot be "nonparametric" methods; something of parametric type is always present, albeit sometimes hidden. Furthermore, the "bootstrap approaches" also mentioned in the Note are unsuitable for stochastic processes as they distort the stochastic structure—particularly in the presence of LRD. Therefore, I suggest making these clarifications and limiting the references to such types of models (as well as to ARMA-type models whose value is only historical, I believe). Instead, I suggest extending the review to other models, more appropriate for hydrological applications, such as those suggested by other commenters.

10. Could the authors double check their equations? Is an imaginary unit missing somewhere in equation (2)? Could they correct the notation in eqn (3)? (Is 'rand' meant to be a subscript?).

11. Finally, I uphold the other commenters in congratulating the authors and I particularly second Panayiotis Dimitriadis in congratulating them for using modest phraseology. I would add in the reasons for congratulation the fact that they do not follow the clichés and fashionable paths: for example they limit their mentions to climate impacts and nonstationarity, a notion that has become a must in hydrological papers—often by authors who do not know what it actually is (see Koutsoyiannis and Montanari 2015; Serinaldi and Kilsby, 2015).

**References**

Brunner, M.I., Bárdossy, A., and Furrer, R. (2019). Technical note: Stochastic simulation of streamflow time series using phase randomization, Hydrol. Earth Syst. Sci. Discuss., doi: 10.5194/hess-2019-142.

Dimitriadis, P., and Koutsoyiannis, D. (2015). Climacogram versus autocovariance and power spectrum in stochastic modelling for Markovian and Hurst–Kolmogorov processes, *Stochastic Environmental Research & Risk Assessment*, 29 (6), 1649–1669, doi: 10.1007/s00477-015-1023-7.

Koutsoyiannis, D. (2016). Generic and parsimonious stochastic modelling for hydrology and beyond, *Hydrological Sciences Journal*, 61 (2), 225–244, doi: 10.1080/02626667.2015.1016950.

Koutsoyiannis, D. (2019). Time's arrow in stochastic characterization and simulation of atmospheric and hydrological processes, *Hydrological Sciences Journal*, doi: 10.1080/02626667.2019.1600700,.

Koutsoyiannis, D., and Montanari, A. (2015). Negligent killing of scientific concepts: the stationarity case, *Hydrological Sciences Journal*, 60 (7-8), 1174–1183, doi: 10.1080/02626667.2014.959959,.

Livadiotis, G., and McComas, D.J. (2013). Understanding kappa distributions: A toolbox for space science and astrophysics. *Space Science Reviews*, 175(1-4), 183-214.

Müller, T., Schütze, M. and Bárdossy, A. (2017). Temporal asymmetry in precipitation time series and its influence on flow simulations in combined sewer systems. *Advances in Water Resources*, 107, 56-64.

Olbert S. (1968) Summary of Experimental Results from M.I.T. Detector on IMP-1. In: Carovillano R.L., McClay J.F., Radoski H.R. (eds) Physics of the Magnetosphere. Astrophysics and Space Science Library (A Series of Books on the Recent Developments of Space Science and of General Geophysics and Astrophysics Published in Connection with the Journal Space Science Reviews), vol 10. Springer, Dordrecht.

Serinaldi, F., and Kilsby, C.G. (2015). Stationarity is undead: Uncertainty dominates the distribution of extremes, *Advances in Water Resources*, 77, 17-36.

---

## Author Comment (AC1) · 21 May 2019

We thank the reviewers and the commentators for acknowledging the value of our work, their feedback, and their constructive comments. We appreciate the wide range of inputs, which allowed us to enrich our introduction and discussion section. In addition to many useful references, the reviewers and commentators point out where and how the stochastic streamflow generator could be reformulated or extended. While each of the points risen is valid, most of them would make the model more complex, less flexible, and less generalizable. With the stochastic simulator presented in the manuscript, we aim at proposing a simple and flexible tool, which can be adapted to different contexts.

[Figure]

This is facilitated by the provision of the simulation procedure as an R-package. In order to guarantee flexibility and generalizability, we combine phase randomization, which is a nonparametric approach for the generation of a time dependence structure, with the flexible four-parameter kappa distribution. Making the model more complex or more parametric would imply a loss in flexibility. We therefore would like to retain the main features of the model proposed. However, we agree that a more profound discussion of its limitations and potential extensions is necessary and valuable. We also agree that the issue with the replacement of negative values by zero values needs to be addressed and that the use of an empirical marginal distribution can in some cases be sufficient. In the PDF attached, we address the points risen by the two reviewers Ashish Sharma and Demetris Koutsoyiannis and state how we would like to address them in a revised version of the manuscript. Our replies to the reviewers' comments are written in blue and italic to distinct them from the reviewers' comments.

On the behalf of all co-authors, Yours sincerely,

Manuela Brunner

Please also note the supplement to this comment:
https://www.hydrol-earth-syst-sci-discuss.net/hess-2019-142/hess-2019-142-AC1-supplement.pdf

[Figure]

**Supplement:**

**Dear Dr. Peleg,**

We thank the reviewers and the commentators for acknowledging the value of our work, their feedback, and their constructive comments. We appreciate the wide range of inputs, which allowed us to enrich our introduction and discussion section. In addition to many useful references, the reviewers and commentators point out where and how the stochastic streamflow generator could be reformulated or extended. While each of the points risen is valid, most of them would make the model more complex, less flexible, and less generalizable. With the stochastic simulator presented in the manuscript, we aim at proposing a simple and flexible tool, which can be adapted to different contexts. This is facilitated by the provision of the simulation procedure as an R-package. In order to guarantee flexibility and generalizability, we combine phase randomization, which is a nonparametric approach for the generation of a time dependence structure, with the flexible four-parameter kappa distribution. Making the model more complex or more parametric would imply a loss in flexibility. We therefore would like to retain the main features of the model proposed. However, we agree that a more profound discussion of its limitations and potential extensions is necessary and valuable. We also agree that the issue with the replacement of negative values by zero values needs to be addressed and that the use of an empirical marginal distribution can in some cases be sufficient.

Below, we address the points risen by the two reviewers Ashish Sharma and Demetris Koutsoyiannis and state how we would like to address them in a revised version of the manuscript. Our replies to the reviewers' comments are written in blue and italic to distinct them from the reviewers' comments.

On the behalf of all co-authors,

Yours sincerely,

Manuela Brunner

**Reviewer 1 (Ashish Sharma)**

My congratulations to the authors on this excellent paper. Very glad to see a clever adopted to frequency domain alternatives in formulating a stochastic streamflow generator. My comments below are aimed to enhance the presentation and I am in support of publication once these have been addressed. Comments are:
**Reply:** *Thank you for acknowledging the value of our work and for the constructive comments, which help to enrich the introduction and discussion section.*

line 2/9 - The authors are missing the works by Keylock (10.1029/2012WR011923). This work performed resampling to an existing time series using phase randomization in the frequency domain. If I remember correctly, it had some nice inclusion of ICA to tackle the multivariate issue, and wavelets to get around nonstationarity in the data that cannot be handled using a fourier transformation alone. I think they need to read those papers (I am familiar with the above one but there may be more since) and acknowledge them here, and also try and show how their work distinguishes itself from the above paper.
**Reply:** *The work by Keylock (2007) indeed shows many parallels to the approach presented in this paper. His approach is not directly based on the Fourier transformation but rather based on the wavelet decomposition of a signal. Instead of the phases of the Fourier transform, the wavelet coefficients are (partly) randomized. The randomized series are then backtransformed to the time domain by using a rank-ordering procedure as*

*presented in the approach used in our manuscript. Keylock (2012) later extended the procedure to the joint simulation at multiple sites. The work by Keylock will be acknowledged in the introduction and discussion section.*

line 3/21: I think the work by Mehrotra (10.1029/2005JD006637) should be acknowledged here as it represents essentially something analogous to a ARMAX type of a model even though it is cast as a stochastic downscaling approach. A mention should be made on the ability to preserve low frequency variability, which I believe the proposed approach will be able to address as well.
**Reply:** *The work by Mehrotra and Sharma (2006) will be acknowledged as an approach allowing for the extension of Markov chains to multiple sites by using spatially correlated random numbers.*

Line 3/35: Even though it relates to the problem of correcting systematic biases, given the use of phase transformation (not randomisation), the approaches of Nguyen should perhaps be acknowledged for completeness. The rationale behind these approaches and the one here has a lot in common. (10.1007/s00382-018-4191-6, 10.1016/j.jhydrol.2016.04.018).
**Reply:** *Thank you for pointing out these references. We will acknowledge the work of Nguyen et al. (2019) in the discussion section where we talk about options of how to improve the representation of the cross-correlation in simulated series.*

line 5/21: The authors may want to look through the details of (10.1007/s00382-018-4191-6, 10.1016/j.jhydrol.2016.04.018) as they performed another level of preprocessing - they fit a Thomas Feiring type model to the monthly data and after that structure was removed, the Fourier transformation was performed. This was done after trying with the steps referred to above, as it was found to exhibit clear advantages.
**Reply:** *We experimented with different types of deseasonalization techniques and found that the normalization at daily scale served the purpose of removing seasonality in the data well. Compared to using a Thomas-Fiering model, the approach used here is non-parametric and does not assume any temporal seasonality structure. Deseasonalizing by a Thomas-Fiering model and re-adding this seasonality at the end, might be valuable if the reproduction of the lag-1 autocorrelation was an issue, which was not the case here. However, it requires the fitting of a parametric model which is data dependent. Our routine works independent of the time resolution of the data and is easily adjustable to different contexts. We show that the ACF of the observed data is nicely preserved by the approach employed in our study.*

line 6/21: Setting negatives to zero is not a clean option. Please refer to the Keylock paper above again on how they restricted their approach to resampling to avoid having to set negatives to zero.
**Reply:** *We agree that setting negative values to zero is indeed not very elegant. We will change the algorithm in order to avoid this. Instead of replacing negative values by zero, we will replace these values by a value sampled from a uniform distribution in the interval [0, min(Q_obs_day)], where min(Q_obs_day) represents the minimum of the observed values corresponding to the day under consideration.*

line 11/10: Underestimation of cross-correlations is I think addressed well in (10.1007/s00382-018-4191-6). The trick that is used is to not randomly generate phases for all variables, but for a "key" variable (say biggest streamflow mean location). And then maintain the phase difference between alternate sites. The phase difference in space helps capture the cross-dependence attributes.
**Reply:** *The approach proposed by Nguyen et al. (2019) for a good representation of the cross-correlation between two or multiple time series in the context of bias correction could also be adopted in the stochastic simulation framework presented in our manuscript.*

*The discussion section will be extended by the phase-difference correction functions introduced by* Nguyen et al. (2019)*.*

Lastly, I feel not addressing the issue of non-stationarity in a stochastic generation paper under our present climate should be discouraged. The issue of nonstationarity can be addressed in the sense of a discussion by thinking of adding an exogenous predictor variable set in the formulation, which can impart the changes needed. Some discussion to that effect would be good to include in the paper before it is published.

**Reply:** *We agree that addressing non-stationarity, if present, is important. The manuscript therefore contains a note stating that the stochastic generator could be applied using discharge time series simulated with a hydrological model driven by meteorological data simulated with a GCM (and RCM) (p 13. L20-22 in the original manuscript). We will slightly extend the discussion by discussing more options of how to adjust the phase randomization approach to non-stationary conditions.*

**Reviewer 2 (Demetris Koutsoyiannis)**

1. The Technical Note by Brunner et al. (2019) implements a useful idea for easy stochastic simulation of daily streamflow, based on spectral representation and phase randomization. The method has several limitations (see below) but it is practical and useful, and it certainly deserves publication. I believe several issues can be improved before final publication and therefore I am providing some suggestions. I also appreciate the commentaries by Francesco Serinaldi, Ioannis Tsoukalas and Panayiotis Dimitriadis, who provided a lot of information to the authors. I think this information is useful to optimize their Note and also to put it in the context of modern and older literature, some of which is missing in the literature review. I believe that not everything suggested in the commentaries needs to be addressed, as this would change the orientation of the Note. However, with several changes in the formulations and a few expansions, rather than additional analyses, the Note could be improved. My own suggestions, which I list in the following points, fall in two categories: (a) recognition of the limitations of the method and (b) improvements in formulations, phraseology and terminology.

**Reply:** *Thank you for acknowledging the usefulness and practicality of our approach. We would like to retain the main characteristics of the stochastic simulation procedure proposed here, which make it a flexible and generalizable tool. However, we will extend the discussion section in order to discuss its limitations more in depth.*

2. A first limitation, which in the current version is not stated clearly, is the severe dependence of the method on the sample size of observations. The synthetic series has the same length as the observed series. The authors properly recognize the importance of respecting long-range dependence (LRD) in simulation. However, to study its effect in hydrosystems we need synthetic series much longer than the observed. The use of ensembles of small-length time series may not be equivalent with using a long time series as each member of the ensemble is independent from the others.

**Reply:** *Thank you for pointing out the need to address this limitation. We agree that producing an ensemble of time series of the same length as the observed one might not be equivalent to the generation of one very long time series if long-range dependence features are present which exceed the length of the observed series. However, such features cannot be generated anyway since the model is fitted based on a limited number of years of observations. We added this issue to the discussion.*

3. A second limitation is the absence of a model for time dependence. While the authors correctly adopt a model for the marginal distribution (e.g. they state "Using the empirical distribution instead of the Kappa distribution would prevent us from obtaining values that go beyond the range of observed data…") their method misses to do so for the dependence structure. The empirical autocorrelogram and periodogram are affected by significant bias and huge noise (see references provided by Panayiotis Dimitriadis) and if we do not use a model, then we reproduce a particular random realization, in terms of autocorrelogram and periodogram, in all our simulated series. I believe authors' statement "The periodogram, the empirical counterpart of the power spectrum, shows high values at those frequencies which correspond to strong periodic components" is only partly true and perhaps misleading. The periodogram could be regarded a realization of a stochastic process per se (on the frequency domain) and its peaks do not necessarily reflect a real peak in the "true" power spectrum. The same thing happens with the autocorrelogram. For example, the ups and downs in the empirical autocorrelograms in Fig. 5 may well be sampling artefacts, which we do not need to reproduce—but the method does reproduce them. If the authors have difficulty to accept my comment, I would suggest doing an experiment with a particular (smooth) autocorrelation function and see the ups and downs in the produced autocorrelogram and periodogram of a single realization.

**Reply:** *As correctly stated above, the approach brought forward here employs a nonparametric approach for the stochastic generation of the time dependence structure. In theory, one could try to find a suitable parametric model to represent this time dependence. We are convinced, however, that doing so would be far from straightforward since it would be very difficult to represent the complexity in the time dependence structure at different time scales ranging from short to long range. Furthermore, the choice of a model would be dependent on the catchment area. Besides its flexibility in reproducing dependence structures at different ranges, the approach presented here has the advantage of being applicable to any dataset of interest without having to fit a parametric time dependence model. The nonparametric time dependence model is flexible, applicable in any catchment, and easy to apply.*

4. A third limitation is the lack of parsimony of the entire methodology. From the statement "We fit a separate distribution for each day to take into account seasonal differences in the distribution of daily streamflow values" one can imagine that the overall method encompasses lots of parameters. Apparently, it is nowadays easy to do calculations with lots of parameters but, in my view, stochastics goes beyond calculations and algorithms. Parsimony in stochastic modelling is always important (see Koutsoyiannis 2016).

**Reply:** *We agree that the stochastic model presented in our study involves a fair amount of parameters. However, this is necessary if we would like to achieve a proper representation of the distributions of the daily discharge values in addition to being able to generate values outside the range of the observed values. If the user is satisfied with values within the observed values, he/she might forgo the use of a theoretical distribution and use the empirical distribution of the observed values for backtransformation instead. This option will be implemented in the new version of the R-package PRsim and the issue will be addressed in the revised version of the manuscript.*

5. A final limitation for the particular time scale of modelling, i.e. daily, is the lack of explicit modelling of time irreversibility (an issue also mentioned in the comment by Francesco Serinaldi). This would not be an issue if the time scale was monthly or longer, but I suspect that it is for the daily scale (see Koutsoyiannis 2019 and also Müller et al. 2017). I clarify here that I do not suggest changing the method to overcome the limitations (e.g. to become more parsimonious or to take irreversibility into account). Rather, I just recommend stating them in a clear and explicit manner.

**Reply:** *Thank you for pointing out the issue of time irreversibility. We agree that this aspect*

*is not explicitly considered in the modeling strategy. Neither is it considered in most existing modeling approaches. The lack of consideration of the time irreversibility issue is mentioned in the discussion section.*

6. Now coming to the second category of my suggestions, I would recommend avoiding the name kappa distribution for the chosen distribution. It is true that in hydrological literature this name is in common use, but if we wish to facilitate communication with other disciplines, we should be aware that the name kappa distribution has another meaning in statistical thermodynamics—namely it is used to describe Cauchy-type (or Student-like) distributions in motion of particles (e.g. Olbert, 1968; Livadiotis and McComas 2013). The specific distribution used in the Note (which I do not think is a generalization of GEV as suggested by Ioannis Tsoukalas), is commonly (in most disciplines) referred to as the Dagum distribution—see https://en.wikipedia.org/wiki/Dagum_distribution. In addition, in terms of sign conventions in eqn. (4), I would suggest changing the signs of k and h and replacing the two minus signs in front of them with plus signs. This will make the expression more convenient and intuitive, and also complying to the standard notation used in other disciplines (e.g. as seen in the above web site).

**Reply:** *Thank you for highlighting that there was some confusion about the use of the term kappa distribution. The development of the name of the kappa distribution introduced by Hosking in 1994 has indeed an interesting history and there is a huge potential for confusion.* Mielke (1973) *introduced a three-parameter kappa distribution.* Hosking (1994) *generalized this distribution to a four-parameter distribution and called it "the four-parameter kappa distribution", which is a generalization of the generalized logistic, GEV, and generalized Pareto distributions. The Dagum distribution is also related to the three-parameter distribution by Mielke (1973) in the sense that it has the same properties but uses a different parameterization* (Kleiber, 2008). *We here used the four-parameter kappa distribution by Hosking, which offers more flexibility compared to the three-parameter distribution. We retain the notation introduced by Hosking (1994) to stress the link to this original publication. We will add a remark to the text highlighting the link between the article by Hosking and the article by Mielke.*

7. The phrase "Stochastically generated time series mimic the characteristics of observed data and represent sets of plausible but as yet *unobserved* streamflow sequences" (my emphasis) may distort the meaning of what stochastic simulation is. It is not a matter of something that is "yet unobserved" but expected to be observed in the future. It is a matter of producing artificial "realizations" from the stochastic model. A model, stochastic or otherwise, is not identical to the real world.

**Reply:** *The corresponding sentence will be rephrased.*

8. The term "deseasonalization" needs to be used with care and clarification; otherwise it may mislead people to think that, by techniques like that used in the Note, we can get rid of seasonality. This, however, is quite difficult—if ever possible. With transformations of the time series, either linear (as in standardizing by mean and variance of each period) or nonlinear (as in fitting a separate distribution for each period, like what is done in this Note), we can only remove the seasonal effect on the marginal distribution, not that of the joint distribution of a cyclostationary stochastic process. (For example, differences in autocorrelation coefficients in different seasons are not removed by techniques such as the above mentioned). Therefore I suggest replacing "deseasonalization" with "deseasonalization of the marginal distribution."

**Reply:** *Thank you for suggesting this clarification. Deseasonalization will be replaced by deseasonalization of the marginal distribution.*

9. The notion of "nonparametric" techniques referred to in the literature review is, in my opinion, problematic when we deal with stochastic processes with time dependence. As opposite to iid statistics, in which the first "i" (independent) is taken for granted, in stochastics there cannot be "nonparametric" methods; something of parametric type is always present, albeit sometimes hidden. Furthermore, the "bootstrap approaches" also mentioned in the Note are unsuitable for stochastic processes as they distort the stochastic structure—particularly in the presence of LRD. Therefore, I suggest making these clarifications and limiting the references to such types of models (as well as to ARMA-type models whose value is only historical, I believe). Instead, I suggest extending the review to other models, more appropriate for hydrological applications, such as those suggested by other commenters.

**Reply:** *The term nonparametric has been used in the literature for certain types of models used for the generation of stochastic time series (Salas and Lee, 2010). We highlight that these approaches are not suitable for the reproduction of long-range dependence (see p:3, l:16-17). The disadvantages of the ARMA models are also clearly stated (see p:3, l:1-4). We will extend the literature review by more advanced models, which allow for more flexible time dependence structures (e.g.* Tsoukalas et al. (2018)*).*

10. Could the authors double check their equations? Is an imaginary unit missing somewhere in equation (2)? Could they correct the notation in eqn (3)? (Is 'rand' meant to be a subscript?).

**Reply:** *We checked the equations and there was indeed an imaginary unit missing (we misspelled j instead of i). The equations will be corrected in the revised version of the manuscript.*

11. Finally, I uphold the other commenters in congratulating the authors and I particularly second Panayiotis Dimitriadis in congratulating them for using modest phraseology. I would add in the reasons for congratulation the fact that they do not follow the cliches and fashionable paths: for example they limit their mentions to climate impacts and nonstationarity, a notion that has become a must in hydrological papers—often by authors who do not know what it actually is (see Koutsoyiannis and Montanari 2015; Serinaldi and Kilsby, 2015).

**Reply:** *Thank you.*

**Selected comments from the comments by Francesco Serinaldi, Ioannis Tsoukalas, and Panayiotis Dimitriadis**

*The comments provided by the three commentators are highly appreciated and will be used to enrich the introduction, methods, and discussion sections. More specifically, we will address several approaches which could be employed to allow for an improved representation of the cross-correlation in simulated time series; we will address the problem of temporal asymmetry; we will add a comment on the SPARTA model by Tsoukalas et al. (2018) to the introduction; we will specify the estimation method used for the estimation of the parameters of the kappa distribution; we tested the Burr type XII (validation results see Figure 1) and generalized Gamma distributions, which were, however, not flexible enough to model the marginal distributions of the daily discharge values and led to unrealistically extreme high flows in the simulations and will therefore not be considered as alternatives to the kappa distribution.*

[Figure]

**Figure 1: Validation plots for the Birse catchment on discharge time series generated using the Burr type XII distribution.**

**References used in the answers to the reviewers**

Hosking, J.R.M., 1994. The four-parameter kappa distribution. IBM J. Res. Dev. 38, 251–258.

Keylock, C.J., 2012. A resampling method for generating synthetic hydrological time series with preservation of cross-correlative structure and higher-order properties. Water Resour. Res. 48, 1–18. https://doi.org/10.1029/2012WR011923

Keylock, C.J., 2007. A wavelet-based method for surrogate data generation. Phys. D Nonlinear Phenom. 225, 219–228. https://doi.org/10.1016/j.physd.2006.10.012

Kleiber, C., 2008. A auide to the Dagum distributions, in: Chotikapanich, D. (Ed.), Modeling Income Distributions and Lorenz Curves. Springer, Berlin-New York, pp. 97–117. https://doi.org/10.1007/978-0-387-72796-7_6

Mehrotra, R., Sharma, A., 2006. A nonparametric stochastic downscaling framework for daily rainfall at multiple locations. J. Geophys. Res. Atmos. 111, 1–16. https://doi.org/10.1029/2005JD006637

Mielke, P.W., 1973. Another family of distributions for describing and analyzing precipitation data. J. Appl. Meteorol. 12, 275–280.

Nguyen, H., Mehrotra, R., Sharma, A., 2019. Correcting systematic biases across multiple atmospheric variables in the frequency domain. Clim. Dyn. 52, 1283–1298. https://doi.org/10.1007/s00382-018-4191-6

Salas, J.D., Lee, T., 2010. Nonparametric simulation of single-site seasonal streamflows. J. Hydrol. Eng. 15, 284–296. https://doi.org/10.1061/(ASCE)HE.1943-5584.0000189

Tsoukalas, I., Efstratiadis, A., Makropoulos, C., 2018. Stochastic periodic autoregressive to anything (SPARTA) modeling and simulation of cyclostationary processes with arbitrary marginal distributions. Water Resour. Res. 54, 161–185. https://doi.org/10.1111/j.1752-1688.1969.tb04897.x

---

## Referee Comment (RC3) · Simon Michael Papalexiou (Referee) · 23 May 2019

This is a useful and interesting technical note on simulating stream flow time series preserving observed characteristics. Several techniques exist to approximate time series such as preserving moments, using marginal-back transformations, bootstrap, amplitude adjusted Fourier transformations methods, etc. All of them have advantages and disadvantages. This technical note is well-structured, well-written, and the real-world case is nicely demonstrated. The authors' intention to provide an easy-to-apply solution is clear, and although this is a technical note, the added value against previews works on the amplitude adjusted Fourier transformation method should be better high-

lighted in order to strengthen the publication.

Before providing a detailed review, I'm expressing my gratitude to the commenters referring to my 2018 work. Clarifying, I worked on this framework much earlier in 2009 for a multivariate and cyclostationary simulation of daily rainfall (13 stations in Greece) aiming to preserve marginals (the Burr type XII was used), correlations and intermittency. The method was described in detail in a document (Papalexiou, 2010) that includes also mixed-type distributions inflated at arbitrary points and not just at zero. The extended work was published in arXiv (Papalexiou, 2017), followed by the journal publication some months later (Papalexiou, 2018). The scheme also applied in a stationary/nonstationary disaggregation framework preserving marginals and correlations (DiPMaC) (Papalexiou et al., 2018). It was an attempt to provide a simple framework for univariate and multivariate modeling preserving continuous, discrete, binary or mixed-type marginals having positive definite autocorrelation structure (including long memory). The focus was specifically on hydroclimatic variables such as precipitation, streamflow, wind, humidity, etc.

Yet as the saying goes, most things have been already found before; indeed, the first attempts in other scientific fields to simulate time series preserving marginals and correlations using inflated correlations date back in the works of Conner and Hammond (1972), and probably much earlier. A clear presentation for continuous marginals using AR models was given by Li & Hammond in (1975). The authors may wish to check also an old paper entitled "Generation of random signals with specified probability density function and power density spectra" by Gugar and Kavanagh published in 1968. Following Li & Hammond (1975) several papers got published in different fields (e.g., Cario & Nelson, 1997, 1998; Kugiumtzis, 2002; Macke et al., 2009; Demirtas, 2014, 2017; Emrich & Piedmonte, 1991; Macke et al., 2009 to mentioned a few) dealing with several cases. These and many other interesting works yet were not suitable for simulating intermittent processes, like precipitation or stream flows, and most of them suggested demanding iterative procedures to estimate the parent Gaussian correlations

or the cross-correlation upper limits (Papalexiou, 2018). In few papers the same link between the gaussian correlation and the bivariate copula with prescribed marginals is also established (even earlier) by using the same technique, i.e., the fundamental 2-dimensional theorem of the expected value of a transformed random variable (Nataf, 1962; Liu & Der Kiureghian, 1986; H. Li et al., 2008; Lebrun & Dutfoy, 2009; Xiao, 2014). This link can also be established by using the Jacobian of the transformation (Papalexiou, 2018, Eq. 7).

The authors here focus on another method, i.e., the amplitude adjusted Fourier transformation, with the same intent and aiming to generate consistent stream flow time series. They can also see a simulation of daily stream flow preserving a heavy tailed marginal (Burr type III) and a slowly decaying autocorrelation structure in Section 4.3 in Papalexiou (2018). Specific comments that authors may find useful to improve the manuscript are:

1. The authors write: "We hereafter refer to such methods, which are also known as amplitude-adjusted Fourier transformations (Lancaster et al., 2018), as phase randomization simulations. In hydrology, phase randomization simulation has rarely been applied for purposes other than hypothesis testing (Fleming et al., 2002), even though it has desirable properties which make it suitable for a wider range of applications. (...) However, its application is limited to Gaussian data. We here propose the use of phase randomization simulation for the stochastic generation of streamflow time series at individual and multiple sites. To allow for non-Gaussian distributions, as commonly observed for daily streamflow values, we combine the data simulated by phase randomization with the Kappa distribution," Probably missing something here, but amplitude-adjusted Fourier transformations can account for non-Gaussian amplitude distributions. The method proposed by Prichard and Theiler (1994) (authors cite indeed this paper) accounts for non-Gaussian marginal distributions as stated by the authors at page 953 "We account for non-Gaussian amplitude distribution by using the amplitude adjusting algorithm described in in Ref. [8] for each component". Prichard

and Theiler (1994) use the algorithm described in Sec 2.4.3 of Theiler et al (1992), which reads as follows "The idea is to first rescale the values in the original time series so they are gaussian. Then the FF or WFT algorithm can be used to make surrogate time series which have the same Fourier spectrum as the rescaled data. Finally, the gaussian surrogate is then rescaled back to have the amplitude distribution as the original time series.". These techniques (Amplitude adjusted Fourier transformations) are known to preserve the linear correlations of the parent Gaussian process rather than those of the target. Indeed, they were further refined as Iterative Amplitude adjusted Fourier transformations in order to match ACF and marginal distributions of the target variables as closely as possible (Kugiumtzis, 1999; Schreiber & Schmitz, 1996; e.g., Serinaldi & Lombardo, 2017; Venema et al., 2006). So, if not missing something here I see that the methodology proposed in this technical note is related to the procedure applied by Prichard and Theiler (1994) in their second example with a difference spotted in the rescaling of the marginal distribution, where empirical CDF is replaced by a parametric Kappa distribution. Thus, maybe the statement that AAFT cannot deal with non-Gaussian marginals should be revised as my understanding is that AAFT it was devised to deal exactly with this. Of course, replacing empirical CDF with Kappa is more appropriate for stream flow simulations.

2. An important point regards the underestimation of cross-correlation reported by the authors (it could be also observed also in the autocorrelation). It was proven mathematically long time ago (Kendall & Stuart, 1979, p. 600) (Embrechts et al., 2002) that any nonlinear transformation of a gaussian time series reduces the strength of the linear correlations as expressed by the Pearson correlation coefficient. Obviously, this does not affect the rank correlations. Since the authors are not calculating the inflated autocorrelations (or inflated spectrum) it is expected the generated time series to have lower correlations. However, in practice this depends on the transformation used. If the target marginal is bell-shaped then typically it has a small effect in reducing the autocorrelation, yet the effect can be very large for j-shaped target marginals with heavy tails and zero inflated (see Fig. 1 and Fig. 2 and the simulation examples in Fig4-Fig7

in Papalexiou (2018) and the discussion therein). Therefore, the fact that the authors don't observe large differences in the autocorrelation of the simulated time series is case specific and not generally true. This point is important and should be mentioned as there are streamflow series highly inflated at zero and highly skewed; without using inflated correlations for the parent gaussian process it is certain that the transformed series will not match the target process.

3. As previously mentioned early approaches in simulating nongaussian timeseries preserving marginals and correlations date back long time ago (e.g. S. T. Li & Hammond, 1975), yet this framework as it was formulated didn't include the modeling of mixed-type marginals (see Papalexiou 2018) which allows easy simulation of intermittent processes such as precipitation or stemflows of ephemeral streams. To increase the novelty of the paper the authors can easily include mixed-type quantiles (see Eq. 17 in Papalexiou 2018) with the kappa distribution. As far as I know this approach has not been implemented in amplitude adjusted Fourier transformations.

Minor points (p = page, L = Line)

p2L33-p3L4: We can easily use large order AR models to simulate time series having long memory or any other strong autocorrelation structure. Especially AR models have an analytical solution (Yule-Walker system) and the fit and the application e.g., of an AR of order 10000 is a matter of less than a second. This means that we can reproduce exactly the autocorrelation structure up to order p. It should be clear that fitting an AR of any large order to a long memory autocorrelation structure is parsimonious and efficient as all the AR parameters are analytically and without uncertainty estimated by the autocorrelation structure, e.g., if an AR of order 10000 is fitted to an fGn correlation then it is an one-parameter model and not a 10000 parameter model. More details can be found in Papalexiou (2018), where the authors can also see examples of long memory process simulations using AR models and preserving marginals. Also long memory can be approximated by the sum of independent AR(1) processes as suggested by Mandelbrot (1971). So, it is a matter of how these models are applied and definitely

they can reproduce long memory or any other autocorrelation structure.

p4L6-p4L9: The Kappa distribution is a well-established distribution in hydrology and since Hosking (1994) introduced the four-parameter version it has been applied countless times (e.g., Hanson & Vogel, ; Kjeldsen et al., 2017; Park et al., 2009) ). Its importance and flexibility stems for the fact that generalizes important distribution such as the GP, GEV, GLO etc, but it can also be seen as a special case and generalization at the same time of the Burr type XII. Maybe the great disadvantage of the four-parameter version is the location parameter which can end up in supporting a range of values which is inconsistent with the variable under study. For stream flows expected to range in the positive axis this can be problematic. If distributions like the Generalized Gamma or the Burr type XII didn't work, maybe the authors should try with the Burr type III (1 scale, 2 shape pars) (Burr, 1942) or the Generalized beta of the second kind (Mielke & Johnson, 1974) (1 scale, 3 shapes) which has great flexibility and is defined in (0, Infinity). This would solve the issue of negative values or of a lower positive limit but of course the authors may neglect this suggestion.

p5L19 and P6L9-11: To clarify regarding the normal transformation. The authors have fitted the Kappa in each day and then use the Kappa cdf to transform to uniform and then apply the gaussian quantile? Even if they did so the final time series might have normal marginal but the autocorrelation may be different in each week/month or season.

p6L3: The KS-test is not a very robust test. It will not change anything to the analysis, but maybe more robust tests should be used and promoted, e.g., the Anderson-Darling. If it's not much of a trouble the authors could test the fit based on the AD test.

p6L20: I might be missing something here but why is 0 the lower bound of the four-parameter Kappa?

Summarizing, this is well-written and useful technical note that deserves publication after some amendments and literature updates.

Regards,

Simon Michael Papalexiou

References

1. Burr, I. W. (1942). Cumulative Frequency Functions. The Annals of Mathematical Statistics, 13(2), 215–232.

2. Cario, M. C., & Nelson, B. L. (1997). Modeling and generating random vectors with arbitrary marginal distributions and correlation matrix. Technical Report, Department of Industrial Engineering and Management Sciences, Northwestern University, Evanston, Illinois.

3. Cario, M. C., & Nelson, B. L. (1998). Numerical methods for fitting and simulating autoregressive-to-anything processes. INFORMS Journal on Computing, 10(1), 72–81.

4. Demirtas, H. (2014). Joint generation of binary and nonnormal continuous data. Journal of Biometrics & Biostatistics, (S12), 1.

5. Demirtas, H. (2017). Concurrent generation of binary and nonnormal continuous data through fifth-order power polynomials. Communications in Statistics-Simulation and Computation, 46(1), 344–357.

6. Embrechts, P., McNeil, A., & Straumann, D. (2002). Correlation and dependence in risk management: properties and pitfalls. Risk Management: Value at Risk and Beyond, 176223.

7. Emrich, L. J., & Piedmonte, M. R. (1991). A method for generating high-dimensional multivariate binary variates. The American Statistician, 45(4), 302–304.

8. Hanson Lars S., & Vogel Richard. (n.d.). The Probability Distribution of Daily Rainfall in the United States. World Environmental and Water Resources Congress 2008, 1–10. https://doi.org/10.1061/40976(316)585

9.  Hosking, J. R. M. (1994). The four-parameter kappa distribution. IBM Journal of Research and Development, 38(3), 251–258. https://doi.org/10.1147/rd.383.0251

10.  Kendall, M., & Stuart, A. (1979). Handbook of Statistics. Griffin & Company, London.

11. Kjeldsen, T. R., Ahn, H., & Prosdocimi, I. (2017). On the use of a four-parameter kappa distribution in regional frequency analysis. Hydrological Sciences Journal, 62(9), 1354–1363. https://doi.org/10.1080/02626667.2017.1335400

12.  Kugiumtzis, D. (1999). Test your surrogate data before you test for nonlinearity. Physical Review. E, Statistical Physics, Plasmas, Fluids, and Related Interdisciplinary Topics, 60(3), 2808–2816.

13.  Kugiumtzis, D. (2002). Statically transformed autoregressive process and surrogate data test for nonlinearity. Physical Review E, 66(2), 025201.

14.  Lebrun, R., & Dutfoy, A. (2009). An innovating analysis of the Nataf transformation from the copula viewpoint. Probabilistic Engineering Mechanics, 24(3), 312–320. https://doi.org/10.1016/j.probengmech.2008.08.001

15. Li, H., Lü, Z., & Yuan, X. (2008). Nataf transformation based point estimate method. Chinese Science Bulletin, 53(17), 2586. https://doi.org/10.1007/s11434-008-0351-0

16.   Li, S. T., & Hammond, J. L. (1975).   Generation of Pseudorandom Numbers with Specified Univariate Distributions and Correlation Coefficients. IEEE Transactions on Systems, Man, and Cybernetics, SMC-5(5), 557–561. https://doi.org/10.1109/TSMC.1975.5408380

17. Liu, P.-L., & Der Kiureghian, A. (1986). Multivariate distribution models with prescribed marginals and covariances. Probabilistic Engineering Mechanics, 1(2), 105–112. https://doi.org/10.1016/0266-8920(86)90033-0

18.  Macke, J. H., Berens, P., Ecker, A. S., Tolias, A. S., & Bethge, M. (2009). Generating spike trains with specified correlation coefficients. Neural Computation, 21(2), 397–423.

19. Mandelbrot, B. B. (1971). A Fast Fractional Gaussian Noise Generator. Water Resour. Res., 7(3), 543–553.

20. Mielke, P. W., & Johnson, E. S. (1974). Some generalized beta distributions of the second kind having desirable application features in hydrology and meteorology. Water Resources Research, 10(2), 223–226. https://doi.org/10.1029/WR010i002p00223

21. Nataf, A. (1962). Statistique mathematique-determination des distributions de probabilites dont les marges sont donnees. COMPTES RENDUS HEBDOMADAIRES DES SEANCES DE L ACADEMIE DES SCIENCES, 255(1), 42.

22. Papalexiou, S.M. (2010). Stochastic modelling demystified. 10.13140/RG.2.2.34889.60008. https://www.researchgate.net/publication/333323778_Stochastic_modelling_demystified

23. Papalexiou, S. M. (2017). A unified theory for exact stochastic modelling of univariate and multivariate processes with continuous, mixed type, or discrete marginal distributions and any correlation structure. ArXiv:1707.06842 [Math, Stat]. Retrieved from http://arxiv.org/abs/1707.06842

24. Papalexiou, S. M. (2018). Unified theory for stochastic modelling of hydroclimatic processes: Preserving marginal distributions, correlation structures, and intermittency. Advances in Water Resources, 115, 234–252. https://doi.org/10.1016/j.advwatres.2018.02.013

25. Papalexiou, S. M., Markonis, Y., Lombardo, F., AghaKouchak, A., & Foufoula‐-Georgiou, E. (2018). Precise Temporal Disaggregation Preserving Marginals and Correlations (DiPMaC) for Stationary and Nonstationary Processes. Water Resources Research. https://doi.org/10.1029/2018WR022726

26. Park, J.-S., Seo, S.-C., & Kim, T. Y. (2009). A kappa distribution with a hydrological

application. Stochastic Environmental Research and Risk Assessment, 23(5), 579–586. https://doi.org/10.1007/s00477-008-0243-5

27. Schreiber, null, & Schmitz, null. (1996). Improved Surrogate Data for Nonlinearity Tests. Physical Review Letters, 77(4), 635–638. https://doi.org/10.1103/PhysRevLett.77.635

28. Serinaldi, F., & Lombardo, F. (2017). General simulation algorithm for autocorrelated binary processes. Physical Review. E, 95(2–1), 023312. https://doi.org/10.1103/PhysRevE.95.023312

29. Venema, V., Meyer, S., García, S. G., Kniffka, A., Simmer, C., Crewell, S., et al. (2006). Surrogate cloud fields generated with the iterative amplitude adapted Fourier transform algorithm. Tellus A, 58(1), 104–120. https://doi.org/10.1111/j.1600-0870.2006.00160.x

30. Xiao, Q. (2014). Evaluating correlation coefficient for Nataf transformation. Probabilistic Engineering Mechanics, 37(Supplement C), 1–6. https://doi.org/10.1016/j.probengmech.2014.03.010

---

## Short Comment (SC5) · 24 May 2019

Dear Simon,

A couple of years ago, I and Dr. Lombardo spent several weeks attempting to simulate binary time series with prescribed linear correlation. When I saw Papalexiou (2018), I thought that if it were available few month before, we would have saved much time.

Leaving aside the seemingly infinite list of past works dealing with non-Gaussian random variables and desired ACF emerging in this review process, now I see that your stuff has been around for a decade before being published. Next time, please publish

in due time and save my time!

By the way, I have played a little bit with CoSMoS, and I have to say that the possibility to play with mixtures is rather interesting. . . in particular I think about a possible extension of e.g. Kumaraswamy autoregressive models to better model doubly (0-1) inflated proportion or percentage data. . . Anyway.

I agree with your comments about AAFT, but from my point of view, the proposed method seems to be only a rebranded Prichard-Theiler's algorithm where a parametric distribution replaces the empirical ecdf; however, all you reviewers are happy with that; so, ubi maior minor cessat.

Mala tempora currunt, sed peiora parantur.

Cheers

F

―――――――――――――――――

---

## Author Comment (AC2) · 25 Jun 2019

**Reviewer 3 (Simon Paplexiou)**

This is a useful and interesting technical note on simulating stream flow time series preserving observed characteristics. Several techniques exist to approximate time series such as preserving moments, using marginal-back transformations, bootstrap, amplitude adjusted Fourier transformations methods, etc. All of them have advantages and disadvantages. This technical note is well-structured, well-written, and the real-world case is nicely demonstrated. The authors' intention to provide an easy-to-apply solution is clear, and although this is a technical note, the added value against previews works on the amplitude adjusted Fourier transformation method should be better highlighted in order to strengthen the publication.

Before providing a detailed review, I'm expressing my gratitude to the commenters referring to my 2018 work. Clarifying, I worked on this framework much earlier in 2009 for a multivariate and cyclostationary simulation of daily rainfall (13 stations in Greece) aiming to preserve marginals (the Burr type XII was used), correlations and intermittency.

The method was described in detail in a document (Papalexiou, 2010) that includes also mixed-type distributions inflated at arbitrary points and not just at zero.

The extended work was published in arXiv (Papalexiou, 2017), followed by the journal publication some months later (Papalexiou, 2018). The scheme also applied in a stationary/nonstationary disaggregation framework preserving marginals and correlations (DiPMaC) (Papalexiou et al., 2018). It was an attempt to provide a simple framework for univariate and multivariate modeling preserving continuous, discrete, binary or mixed-type marginals having positive definite autocorrelation structure (including long memory). The focus was specifically on hydroclimatic variables such as precipitation, streamflow, wind, humidity, etc.

Yet as the saying goes, most things have been already found before; indeed, the first attempts in other scientific fields to simulate time series preserving marginals and correlations using inflated correlations date back in the works of Conner and Hammond (1972), and probably much earlier. A clear presentation for continuous marginals using AR models was given by Li & Hammond in (1975). The authors may wish to check also an old paper entitled "Generation of random signals with specified probability density function and power density spectra" by Gugar and Kavanagh published in 1968.

Following Li & Hammond (1975) several papers got published in different fields (e.g., Cario & Nelson, 1997, 1998; Kugiumtzis, 2002; Macke et al., 2009; Demirtas, 2014, 2017; Emrich & Piedmonte, 1991; Macke et al., 2009 to mentioned a few) dealing with several cases. These and many other interesting works yet were not suitable for simulating intermittent processes, like precipitation or stream flows, and most of them suggested demanding iterative procedures to estimate the parent Gaussian correlations or the cross-correlation upper limits (Papalexiou, 2018). In few papers the same link between the gaussian correlation and the bivariate copula with prescribed marginals is also established (even earlier) by using the same technique, i.e., the fundamental 2-dimensional theorem of the expected value of a transformed random variable (Nataf, 1962; Liu & Der Kiureghian, 1986; H. Li et al., 2008; Lebrun & Dutfoy, 2009; Xiao, 2014). This link can also be established by using the Jacobian of the transformation (Papalexiou, 2018, Eq. 7).

The authors here focus on another method, i.e., the amplitude adjusted Fourier transformation, with the same intent and aiming to generate consistent stream flow time series. They can also see a simulation of daily stream flow preserving a heavy tailed marginal (Burr type III) and a slowly decaying autocorrelation structure in Section 4.3 in Papalexiou (2018). Specific comments that authors may find useful to improve the manuscript are:

1. The authors write: "We hereafter refer to such methods, which are also known as amplitude-adjusted Fourier transformations (Lancaster et al., 2018), as phase randomization simulations. In hydrology, phase randomization simulation has rarely been applied for purposes other than hypothesis testing (Fleming et al., 2002), even though it has desirable properties which make it suitable for a wider range of applications. (. . .) However, its application is limited to Gaussian data. We here propose the use of phase randomization simulation for the stochastic generation of streamflow time series at individual and multiple sites. To allow for non-Gaussian distributions, as commonly observed for daily streamflow values, we combine the data simulated by phase randomization with the Kappa distribution," Probably missing something here, but amplitude-adjusted Fourier transformations can account for non-Gaussian amplitude distributions. The method proposed by Prichard and Theiler (1994) (authors cite indeed this paper) accounts for non-Gaussian marginal distributions as stated by the authors at page 953 "We account for non-Gaussian amplitude distribution by using the amplitude adjusting algorithm described in in Ref. [8] for each component". Prichard and Theiler (1994) use the algorithm described in Sec 2.4.3 of Theiler et al (1992), which reads as follows "The idea is to first rescale the values in the original time series so they are gaussian. Then the FF or WFT algorithm can be used to make surrogate time series which have the same Fourier spectrum as the rescaled data. Finally, the gaussian surrogate is then rescaled back to have the amplitude distribution as the original time series.". These techniques (Amplitude adjusted Fourier transformations) are known to preserve the linear correlations of the parent Gaussian process rather than those of the target. Indeed, they were further refined as Iterative Amplitude adjusted Fourier transformations in order to match ACF and marginal distributions of the target variables as closely as possible (Kugiumtzis, 1999; Schreiber & Schmitz, 1996; e.g., Serinaldi & Lombardo, 2017; Venema et al., 2006). So, if not missing something here I see that the methodology proposed in this technical note is related to the procedure applied by Prichard and Theiler (1994) in their second example with a difference spotted in the rescaling of the marginal distribution, where empirical CDF is replaced by a parametric Kappa distribution. Thus, maybe the statement that AAFT cannot deal with non-Gaussian marginals should be revised as my understanding is that AAFT it was devised to deal exactly with this. Of course, replacing empirical CDF with Kappa is more appropriate for stream flow simulations.

**Reply:** *We agree that the amplitude-adjusted Fourier transformation procedure proposed by Prichard and Theiler (1994) can deal with non-Gaussian data. However, it does usually not allow for the use of parametric distributions in the back-transformation process. We adjusted the text accordingly by acknowledging that amplitude-adjusted Fourier transform allows for the generation of non-Gaussian data. We specified that our approach differs from amplitude-adjusted Fourier transformation by that it allows for the generation of values beyond the values in the empirical distribution.*

2. An important point regards the underestimation of cross-correlation reported by the authors (it could be also observed also in the autocorrelation). It was proven mathematically long time ago (Kendall & Stuart, 1979, p. 600) (Embrechts et al., 2002) that any nonlinear transformation of a gaussian time series reduces the strength of the linear correlations as expressed by the Pearson correlation coefficient. Obviously, this does not affect the rank correlations. Since the authors are not calculating the inflated autocorrelations (or inflated spectrum) it is expected the generated time series to have lower correlations. However, in practice this depends on the transformation used. If the target marginal is bell-shaped then typically it has a small effect in reducing the autocorrelation, yet the effect can be very large for j-shaped target marginals with heavy tails and zero inflated (see Fig. 1 and Fig. 2 and the simulation examples in Fig4-Fig7 in Papalexiou (2018) and the discussion therein). Therefore, the fact that the authors don't observe large differences in the autocorrelation of the simulated time series is case specific and not generally true. This point is important and should be mentioned as there are streamflow series highly inflated at zero and highly skewed; without using inflated correlations for the parent gaussian process it is certain that the transformed series will not match the target process.

*Reply: We agree that the autocorrelation of a stochastically generated time series might not in all cases well reproduced the autocorrelation of observed time series. We add a section to the discussion section, that discusses this issue: "While the reproduction of the temporal dependence was well reproduced here, this is not necessarily the case under all conditions. Embrechts et al., 2010 have shown that any nonlinear transformation of a Gaussian time series, which is done during backtransformation, reduces the strength of the linear correlations in the time series as expressed by Pearson's correlation coefficient. If one is working with heavy-tailed and zero inflated marginals, it can happen that autocorrelations are reduced during backtransformation (Papalexiou, 2018)."*

3. As previously mentioned early approaches in simulating nongaussian timeseries preserving marginals and correlations date back long time ago (e.g. S. T. Li & Hammond, 1975), yet this framework as it was formulated didn't include the modeling of mixed-type marginals (see Papalexiou 2018) which allows easy simulation of intermittent processes such as precipitation or stemflows of ephemeral streams. To increase the novelty of the paper the authors can easily include mixed-type quantiles (see Eq. 17 in Papalexiou 2018) with the kappa distribution. As far as I know this approach has not been implemented in amplitude adjusted Fourier transformations.

*Reply: We agree that the use of mixed-type marginals can be beneficial in certain cases, where the process to simulate from is intermittent. We therefore made PRSim even more flexible by introducing user-defined distributions to be used in the backtransformation process. The software illustrates the functionality with GEV and GB2 distributions. This user-defined distribution can potentially be a mixture distribution. We did not include an example of a mixture distribution in our technical note because the time series chosen for the analysis were not characterized by intermittency. Furthermore, the use of mixtures of a discrete and a continuous part is delicate as we cannot resort on classical/standard definitions of likelihood tests or confidence intervals.*

**Minor points (p = page, L = Line)**

p2L33-p3L4: We can easily use large order AR models to simulate time series having long memory or any other strong autocorrelation structure. Especially AR models have an analytical solution (Yule-Walker system) and the fit and the application e.g., of an AR of order 10000 is a matter of less than a second. This means that we can reproduce exactly the autocorrelation structure up to order p. It should be clear that fitting an AR of any large order to a long memory autocorrelation structure is parsimonious and efficient as all the AR parameters are analytically and without uncertainty estimated by the autocorrelation structure, e.g., if an AR of order 10000 is fitted to an fGn correlation then it is an one-parameter model and not a 10000 parameter model. More details can be found in Papalexiou (2018), where the authors can also see examples of long memory process simulations using AR models and preserving marginals. Also long memory can be approximated by the sum of independent AR(1) processes as suggested by Mandelbrot (1971). So, it is a matter of how these models are applied and definitely they can reproduce long memory or any other autocorrelation structure.

**Reply:** *We agree that it is possible to approximate an arbitrary spectrum with either a large order AR or many AR(1) processes. However, this approach remains an approximation. A long-memory process can be characterized with a polynomial decay of the spectrum. AR processes have an exponential decay. Hence, it will not be possible to generate a «truly» long range process. It is possible to approximate arbitrarily precisely an empirical spectrum. If one observes a spectrum of length n, n AR(1) processes will allow for the approximation of an empirical spectrum. We specified in the introduction that AR models can be used to generate seemingly long-memory processes if a parametric autocorrelation structure is used to fit the data.*

p4L6-p4L9: The Kappa distribution is a well-established distribution in hydrology and since Hosking (1994) introduced the four-parameter version it has been applied countless times (e.g., Hanson & Vogel, ; Kjeldsen et al., 2017; Park et al., 2009) ). Its importance and flexibility stems for the fact that generalizes important distribution such as the GP, GEV, GLO etc, but it can also be seen as a special case and generalization at the same time of the Burr type XII. Maybe the great disadvantage of the four-parameter version is the location parameter which can end up in supporting a range of values which is inconsistent with the variable under study. For stream flows expected to range in the positive axis this can be problematic. If distributions like the Generalized Gamma or the Burr type XII didn't work, maybe the authors should try with the Burr type III (1 scale, 2 shape pars) (Burr, 1942) or the Generalized beta of the second kind (Mielke & Johnson, 1974) (1 scale, 3 shapes) which has great flexibility and is defined in (0, Infinity). This would solve the issue of negative values or of a lower positive limit but of course the authors may neglect this suggestion.

**Reply:** *Thank you for this suggestion. We tested the Generalized beta distribution of the second kind. It seems to be rather flexible and works fine for certain catchments. In other catchments, however, it produces very extreme, and rather implausible high-flow values. As the distribution is defined in the interval [0,Infinity], it solves the problem with the zero values, but introduces a problem with infinity values. While this and other distributions tested (GEV, Burr type XII, generalized Gamma, Wakeby) were not found to be suitable in our application example, they might be appropriate in other cases. We therefore adjusted the PRSim R-package (version > 1.0) to allow for any type of distribution specified by the user in the back-transformation process.*

p5L19 and P6L9-11: To clarify regarding the normal transformation. The authors have fitted the Kappa in each day and then use the Kappa cdf to transform to uniform and then apply the gaussian quantile? Even if they did so the final time series might have normal marginal but the autocorrelation may be different in each week/month or season.

**Reply:** *It is correct that the kappa distribution was fitted to each day individually and that the autocorrelation shows monthly variations.*

p6L3: The KS-test is not a very robust test. It will not change anything to the analysis, but maybe more robust tests should be used and promoted, e.g., the Anderson-Darling. If it's not much of a trouble the authors could test the fit based on the AD test.

**Reply:** *We agree that the KS-test is not very robust and added the AD test as a goodness-of-fit test. The PRSim package now allows for choosing one of the two tests.*

p6L20: I might be missing something here but why is 0 the lower bound of the four parameter Kappa?

**Reply:** *It is correct that the kappa distribution does not have a lower bound. The sentence was rephrased to "Negative simulated values are replaced by 0, because the kappa distribution does not allow for setting a lower bound".*

Summarizing, this is well-written and useful technical note that deserves publication after some amendments and literature updates.

**References**

1. Burr, I. W. (1942). Cumulative Frequency Functions. The Annals of Mathematical Statistics, 13(2), 215–232.

2. Cario, M. C., & Nelson, B. L. (1997). Modeling and generating random vectors with arbitrary marginal distributions and correlation matrix. Technical Report, Department of Industrial Engineering and Management Sciences, Northwestern University, Evanston, Illinois.

3. Cario, M. C., & Nelson, B. L. (1998). Numerical methods for fitting and simulating autoregressive-to-anything processes. INFORMS Journal on Computing, 10(1), 72– 81.

4. Demirtas, H. (2014). Joint generation of binary and nonnormal continuous data. Journal of Biometrics & Biostatistics, (S12), 1.

5. Demirtas, H. (2017). Concurrent generation of binary and nonnormal continuous data through fifth-order power polynomials. Communications in Statistics-Simulation and Computation, 46(1), 344–357.

6. Embrechts, P., McNeil, A., & Straumann, D. (2002). Correlation and dependence in risk management: properties and pitfalls. Risk Management: Value at Risk and Beyond, 176223.

7. Emrich, L. J., & Piedmonte, M. R. (1991). A method for generating high-dimensional multivariate binary variates. The American Statistician, 45(4), 302–304.

8. Hanson Lars S., & Vogel Richard. (n.d.). The Probability Distribution of Daily Rainfall in the United States. World Environmental and Water Resources Congress 2008, 1– 10. https://doi.org/10.1061/40976(316)585

9. Hosking, J. R. M. (1994). The four-parameter kappa distribution. IBM Journal of Research and Development, 38(3), 251–258. https://doi.org/10.1147/rd.383.0251 10. Kendall, M., & Stuart, A. (1979). Handbook of Statistics. Griffin & Company, London.

11. Kjeldsen, T. R., Ahn, H., & Prosdocimi, I. (2017). On the use of a four-parameter kappa distribution in regional frequency analysis. Hydrological Sciences Journal, 62(9), 1354–1363. https://doi.org/10.1080/02626667.2017.1335400 12. Kugiumtzis, D. (1999). Test your surrogate data before you test for nonlinearity. Physical Review. E, Statistical Physics, Plasmas, Fluids, and Related Interdisciplinary Topics, 60(3), 2808–2816.

13. Kugiumtzis, D. (2002). Statically transformed autoregressive process and surrogate data test for nonlinearity. Physical Review E, 66(2), 025201.

14. Lebrun, R., & Dutfoy, A. (2009). An innovating analysis of the Nataf transformation from the copula viewpoint. Probabilistic Engineering Mechanics, 24(3), 312–320. https://doi.org/10.1016/j.probengmech.2008.08.001

15. Li, H., Lü, Z., & Yuan, X. (2008). Nataf transformation based point estimate method. Chinese Science Bulletin, 53(17), 2586. https://doi.org/10.1007/s11434-008-0351-0 16. Li, S. T., & Hammond, J. L. (1975). Generation of Pseudorandom Numbers with Specified Univariate Distributions and Correlation Coefficients. IEEE Transactions on Systems, Man, and Cybernetics, SMC-5(5), 557–561. https://doi.org/10.1109/TSMC.1975.5408380

17. Liu, P.-L., & Der Kiureghian, A. (1986). Multivariate distribution models with prescribed marginals and covariances. Probabilistic Engineering Mechanics, 1(2), 105– 112. https://doi.org/10.1016/0266-8920(86)90033-0

18. Macke, J. H., Berens, P., Ecker, A. S., Tolias, A. S., & Bethge, M. (2009). Generating spike trains with specified correlation coefficients. Neural Computation, 21(2), 397–423.

19. Mandelbrot, B. B. (1971). A Fast Fractional Gaussian Noise Generator. Water Resour. Res., 7(3), 543–553.

20. Mielke, P. W., & Johnson, E. S. (1974). Some generalized beta distributions of the second kind having desirable application features in hydrology and meteorology. Water Resources Research, 10(2), 223–226. https://doi.org/10.1029/WR010i002p00223 21. Nataf, A. (1962). Statistique mathematique-determination des distributions de probabilites dont les marges sont donnees. COMPTES RENDUS HEBDOMADAIRES DES SEANCES DE L ACADEMIE DES SCIENCES, 255(1), 42.

22. Papalexiou, S.M. (2010). Stochastic modelling demystified. 10.13140/RG.2.2.34889.60008.https://www.researchgate.net/publication/333323778_Stochastic_modelling_demystified 23. Papalexiou, S. M. (2017). A unified theory for exact stochastic modelling of univariate and multivariate processes with continuous, mixed type, or discrete marginal distributions and any correlation structure. ArXiv:1707.06842 [Math, Stat]. Retrieved from http://arxiv.org/abs/1707.06842

24. Papalexiou, S. M. (2018). Unified theory for stochastic modelling of hydroclimatic processes: Preserving marginal distributions, correlation structures, and intermittency. Advances in Water Resources, 115, 234–252. https://doi.org/10.1016/j.advwatres.2018.02.013

25. Papalexiou, S. M., Markonis, Y., Lombardo, F., AghaKouchak, A., & Foufoulaâˇ ARˇ Georgiou, E. (2018). Precise Temporal Disaggregation Preserving Marginals and Correlations (DiPMaC) for Stationary and Nonstationary Processes. Water Resources Research. https://doi.org/10.1029/2018WR022726

26. Park, J.-S., Seo, S.-C., & Kim, T. Y. (2009). A kappa distribution with a hydrological application. Stochastic Environmental Research and Risk Assessment, 23(5), 579– 586. https://doi.org/10.1007/s00477-008-0243-5

27. Schreiber, null, & Schmitz, null. (1996). Improved Surrogate Data for Nonlinearity Tests. Physical Review Letters, 77(4), 635–638. https://doi.org/10.1103/PhysRevLett.77.635

28. Serinaldi, F., & Lombardo, F. (2017). General simulation algorithm for autocorrelated binary processes. Physical Review. E, 95(2–1), 023312. https://doi.org/10.1103/PhysRevE.95.023312

29. Venema, V., Meyer, S., García, S. G., Kniffka, A., Simmer, C., Crewell, S., et al. (2006). Surrogate cloud fields generated with the iterative amplitude adapted Fourier transform algorithm. Tellus A, 58(1), 104–120. https://doi.org/10.1111/j.16000870.2006.00160.x

30. Xiao, Q. (2014). Evaluating correlation coefficient for Nataf transformation.  Probabilistic Engineering Mechanics, 37(Supplement C), 1–6. https://doi.org/10.1016/j.probengmech.2014.03.010

**References used in the answers to the reviewers**

Embrechts, P., McNeil, A.J., Straumann, D., 2010. Correlation and dependence in risk management: Properties and pitfalls, in: Dempster, M.A.H. (Ed.), Risk Management. Cambridge University Press, Cambridge, pp. 176–223. https://doi.org/10.1017/cbo9780511615337.008

Hosking, J.R.M., 1994. The four-parameter kappa distribution. IBM J. Res. Dev. 38, 251–258.

Keylock, C.J., 2012. A resampling method for generating synthetic hydrological time series with preservation of cross-correlative structure and higher-order properties. Water Resour. Res. 48, 1–18. https://doi.org/10.1029/2012WR011923

Keylock, C.J., 2007. A wavelet-based method for surrogate data generation. Phys. D Nonlinear Phenom. 225, 219–228. https://doi.org/10.1016/j.physd.2006.10.012

Kleiber, C., 2008. A auide to the Dagum distributions, in: Chotikapanich, D. (Ed.), Modeling Income Distributions and Lorenz Curves. Springer, Berlin-New York, pp. 97–117. https://doi.org/10.1007/978-0-387-72796-7_6

Mehrotra, R., Sharma, A., 2006. A nonparametric stochastic downscaling framework for daily rainfall at multiple locations. J. Geophys. Res. Atmos. 111, 1–16. https://doi.org/10.1029/2005JD006637

Mielke, P.W., 1973. Another family of distributions for describing and analyzing precipitation data. J. Appl. Meteorol. 12, 275–280.

Nguyen, H., Mehrotra, R., Sharma, A., 2019. Correcting systematic biases across multiple atmospheric variables in the frequency domain. Clim. Dyn. 52, 1283–1298. https://doi.org/10.1007/s00382-018-4191-6

Papalexiou, S.M., 2018. Unified theory for stochastic modelling of hydroclimatic processes: Preserving marginal distributions, correlation structures, and intermittency. Adv. Water Resour.

115, 234–252. https://doi.org/10.1016/j.advwatres.2018.02.013

Salas, J.D., Lee, T., 2010. Nonparametric simulation of single-site seasonal streamflows. J. Hydrol. Eng. 15, 284–296. https://doi.org/10.1061/(ASCE)HE.1943-5584.0000189

Tsoukalas, I., Efstratiadis, A., Makropoulos, C., 2018. Stochastic periodic autoregressive to anything (SPARTA) modeling and simulation of cyclostationary processes with arbitrary marginal distributions. Water Resour. Res. 54, 161–185. https://doi.org/10.1111/j.1752-1688.1969.tb04897.x